# Orbital modulation of subtropical versus subantarctic moisture sources in the southeast Pacific mid-latitudes

Jérôme Kaiser [1] ✉, Enno Schefuß [2], James Collins[3,9], René Garreaud[4,5], Jan-Berend W. Stuut [2,6,7], Nicoletta Ruggieri[3], Ricardo De Pol-Holz [8] & Frank Lamy [3]

Reconstructing rainfall variability and moisture sources is a critical aspect to understand past and future hydroclimate dynamics. Here, we use changes in the deuterium content of land-plant leaf waxes from two marine sediment cores located off Chile to reconstruct changes in rainfall amount and variation in moisture sources over the last ~50 ka. The records indicate increased moisture in central Chile during precession maxima, but an obliquity modulation is evident in southern Chile. While the southern westerly winds are the dominant factor of precipitation in southern Chile by bringing moisture and perturbations from the extratropics, the subtropics represent an additional moisture source during precession maxima due to a stronger subtropical jet increasing moisture transport from the tropics to the mid-latitudes. These findings imply that a combination of orbital modulation of moisture sources and rainfall amount explains the last glacial moisture maximum and early Holocene moisture minimum in south-central Chile.

The Southern Hemisphere westerly wind belt (SWW) plays an important role in mid- and high-latitude atmosphere–ocean circulation. The strong seasonal latitudinal shifts and year-round influence of the SWW along the Chilean continental margin create a pronounced rainfall gradient with annual values ranging from 100 mm at 27° S to 2000 mm around 45° S[1,2]. In austral summer, the southeast Pacific anticyclone expands poleward, preventing the equatorward progression of cold fronts and restricting rainfall to southern Chile (37–45° S). During austral winter, the SWW influence extends substantially northward and occasionally extra-tropical cyclones reach central Chile (30–36° S) as the southeast Pacific anticyclone weakens and retreats equatorward[3–6]. In the southeast Pacific, the strength and latitudinal position of the austral winter SWW are controlled by the upper level Pacific subtropical jet stream (STJ). The STJ is particularly well developed over the

Pacific sector of the SWW[6] as this sector is characterized by a split jet with a strong STF and a weaker subpolar jet[5,7] that is dynamically linked to tropical sea-surface temperature (SST) changes[8].

Direct proxy evidence of SWW variability in the past remains difficult to obtain. Most continental and marine SWW reconstructions traditionally focus on the last glacial maximum (LGM) and overall indicate a northward shift and/or intensification of the SWW[9] that is accompanied by a northward displacement of the Southern Ocean oceanographic fronts. A similar trend is also evident for Holocene changes in the strength and position of the SWW along Chile[10,11].

A particularly well-suited location to study rainfall changes is the western side of southern South America where SWW position and strength control almost all continental rainfall. Such SWW-controlled precipitation proxy records from central and southern Chile suggest

[1]Leibniz Institute for Baltic Sea Research Warnemünde, Rostock, Germany. [2]MARUM—Center for Marine Environmental Sciences, Bremen University, Bremen, Germany. [3]Alfred-Wegener-Institut Helmholtz-Zentrum für Polar- und Meeresforschung (AWI), Bremerhaven, Germany. [4]Center for Climate and Resilience Research (CR)2, University of Chile, Santiago, Chile. [5]Department of Geophysics, University of Chile, Santiago, Chile. [6]Department of Ocean Systems, NIOZ—Royal Netherlands Institute for Sea Research and Utrecht University, Texel, The Netherlands. [7]Department of Earth Sciences, VU—Vrije Universiteit Amsterdam, Amsterdam, The Netherlands. [8]Centro de Investigación GAIA-Antártica (CIGA), University of Magallanes, Punta Arenas, Chile. [9]Present address: Thermo Fisher Scientific (Bremen) GmbH, Bremen, Germany. ✉e-mail: jerome.kaiser@io-warnemuende.de

that precipitation changes are related primarily to zonally symmetric latitudinal shifts or intensity changes of the SWW at millennial and orbital timescales in the southeast Pacific[12–17]. A one-million-year rainfall record based on continental slope sediments of the southern Atacama Desert provides evidence that the austral winter South Pacific SWW configuration varied zonally asymmetric on precessional timescales (19/23-kyr cycles), namely, a stronger split jet and overall stronger SWW in the South Pacific sector and weaker SWW in the other ocean basins during precession maxima[18]. Based on modelling results, the authors suggested that rainfall changes at these timescales are linked to meridional shifts in water vapour transport from the lower latitudes towards the southern Atacama Desert and reflect a modulation of the split in the austral winter South Pacific STJ. However, direct proxy evidence on potential low- versus high-latitude moisture sources remains elusive.

Sedimentary $n$-alkanes, which are constituents of the epicuticular leaf wax of terrestrial plants[19], represent a unique archive for the hydrogen isotopic composition of past precipitation and, thus, are a proxy for past atmospheric dynamics[20–26]. Leaf wax $\delta D$ ($\delta D_{wax}$) values from sediments are significantly correlated with rainfall $\delta D$ values, in particular in the absence of large-scale vegetation changes[27–32]. Changes in $\delta D_{wax}$ therefore reflect changes in rainfall $\delta D$ due to variability in rainfall intensity and moisture sources[33,34].

In the present study, we use $\delta D_{wax}$ as a moisture source indicator[35]. Besides condensation temperature and isotopic changes due to variable ice mass in the past, which can be accounted for by independent parameters, rainfall $\delta D$ incorporates information on rainfall amounts (amount effect due to limited re-evaporation during high rainfall events) and moisture sources due to different initial isotopic composition (depending on moisture, air temperature and SST of the oceanic source area of the precipitation). These principles enabled the distinction of moisture sources of different origins (Atlantic versus Indian Ocean[36]), but also between proximal from remote oceanic moisture sources in the North Atlantic[26], both related to distinct initial isotope compositions of moisture and differential

rainout history during transport. Here, we perform $\delta D_{wax}$ analyses on two sediment cores from the continental slope of the Chilean margin, GeoB7139-2 at 30° S and ODP1233/GeoB7196-1 at 41° (Fig. 1A), to use the moisture source imprint in $\delta D_{wax}$ and assess the spatio-temporal evolution of moisture sources over the last ~50 ka, which cannot be evaluated by traditional rainfall proxies.

## Results

### Present-day moisture sources

To assess the likely sources of moisture along the present-day Chilean coast, a 4-day back trajectory analysis during 50 rainy days was performed at five coastal meteorological stations (Fig. 1B): La Serena (30° S), Santo Domingo (33° S), Concepcion (37° S), Puerto Montt (41° S) and Tortel (50° S). The results indicate a gradual transition in the source of air parcels that precipitate over the Chilean coast depending on the latitude of the meteorological station. For the subtropical stations (30–33° S), the air parcels originate mainly (75–93%) from the northwest and not far from the coast. In contrast, for the midlatitudes (41–50° S), the air parcels originate also from the northwest (55–63%), but much further away from the coast. The situation at 37° S is intermediate. Due to its shorter transport path, the subtropical moisture should thus be isotopically enriched. Different moisture sources are also distinct in their deuterium excess content with ca. 4‰ higher values at 30° S compared to 41° S as the evaporation degree of water vapour is higher in the warmer, low latitude source areas (Supplementary Table S1). This effect amplifies the isotopic differences of the subtropical versus mid-latitude moisture sources.

### $\delta D_{wax}$ and $\delta^{13}C_{wax}$ variability

Leaf wax deuterium ($\delta D_{wax}$) and carbon ($\delta^{13}C_{wax}$) isotope values of the $n$-$C_{31}$ alkane were measured and used to infer rainfall isotope changes and vegetation sources, respectively (Supplementary Tables S2–S5). The $\delta^{13}C_{wax}$ mean and standard deviation values are $-31.9 \pm 0.3‰$ ($n = 52$) for core GeoB7139-2 (30° S) and $-33.1 \pm 0.2‰$ ($n = 71$) for ODP Site 1233 (ODP1233; 41° S). The $\delta^{13}C_{wax}$ values are very invariant for

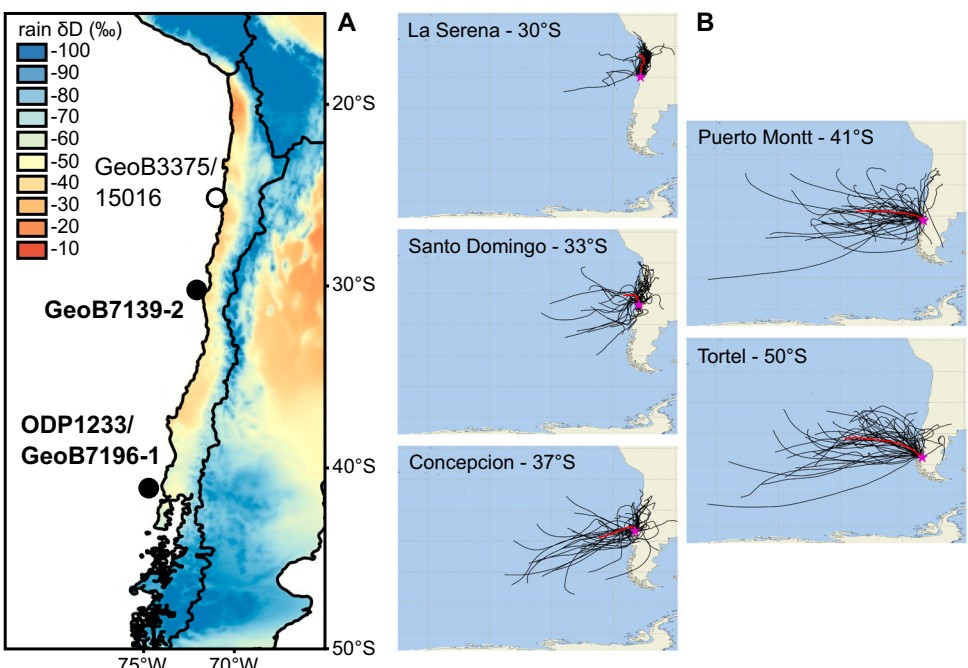

**Fig. 1 | Sediment core locations and trajectory of rain events along the Chilean coast. A** Location of sediment core GeoB7139-2 (30° S) and ODP1233/GeoB7196-1 (41° S) off Chile and spatial distribution of mean annual $\delta D$ values of atmospheric precipitation (www.waterisotopes.org[77]). Core GeoB3375/15016

location[18] is also shown. **B** Back trajectory analysis of rain events at five meteorological stations (stars) along the Chilean coast: La Serena (30° S), Santo Domingo (33° S), Concepcion (37° S), Puerto Montt (41° S) and Tortel (50° S). The bold red lines represent the mean back trajectory for each station.

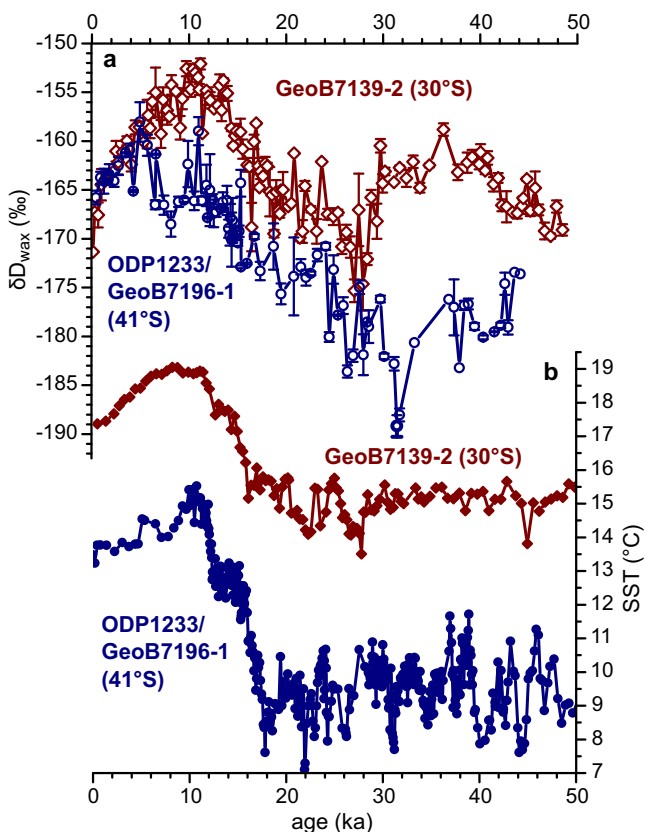

**Fig. 2 | Late glacial to Holocene records off central (30° S) and southern (41° S) Chile. a** $\delta$D records of the n-$C_{31}$ alkane ($\delta$D$_{wax}$) with error bars (standard deviation). **b** Sea-surface temperature (SST) reconstructions. The records are from sediment cores GeoB7139-2 and ODP1233/GeoB7196-1. The SST datasets were published in previous studies[16,74,75].

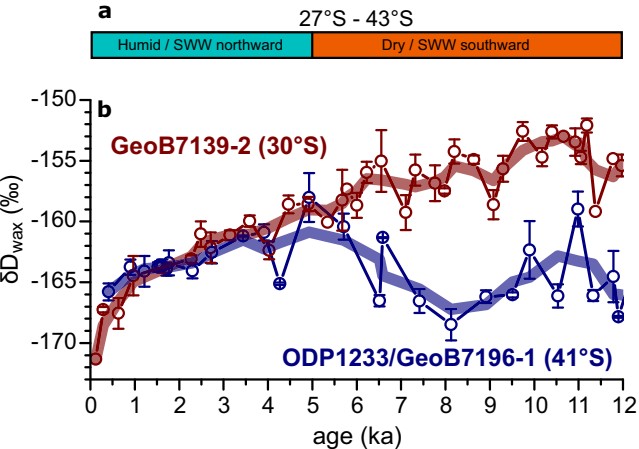

**Fig. 3 | Holocene moisture changes in coastal north-central Chile. a** Synthesis of moisture-related, continental records between 27° S and 43° S[40,45–57]. **b** $\delta$D records of the n-$C_{31}$ alkane ($\delta$D$_{wax}$) off central (GeoB7139-2) and southern (ODP1233/GeoB7196-1) Chile. The error bars (standard deviation) are shown.

both cores (Supplementary Fig. S1). The $\delta$D$_{wax}$ record of core GeoB7139-2 (Fig. 2a) ranges between −175‰ and −152‰ with a mean value of −163 ± 5‰ (n = 123). The $\delta$D$_{wax}$ values are around −166‰ during 50–18 ka, increase by +13‰ until 10 ka (−153‰), and decrease by −18‰ towards the present time (−171‰). For ODP1233/GeoB7196-1 (Fig. 2a), the $\delta$D$_{wax}$ values range between −189‰ and −158‰ with a mean value of −171 ± 7‰ (n = 87). The $\delta$D$_{wax}$ values are ca. −178‰ during 42–26 ka, increase by +20‰ until 5 ka (−158‰), and decrease by −8‰ towards the present time (−166‰).

## Discussion
### Origin of the $\delta$D$_{wax}$ signal
$\delta$D$_{wax}$ values may be affected to a minor extent by biosynthetic effects related to plant life forms[37,38]. The $\delta^{13}$C$_{wax}$ mean values of both cores suggest that $C_3$ plants remain the principal source of the n-$C_{31}$ alkane over the last ~50 ka at 30° S and 41° S (Supplementary Fig. S1). Indeed, as inferred from pollen records the regional vegetation on the adjacent land at the locations of both cores was dominated by $C_3$ plants (trees and woodland) during the entire time interval[39–41]. Only during the deglaciation and the early Holocene grasses and shrubs increased, but $C_4$ monocotyledonous plants remain in low proportion (<20%). Therefore, $\delta$D$_{wax}$ values are not significantly affected by changes in the type of vegetation and the $\delta$D$_{wax}$ records off central and southern Chile are predominantly influenced by atmospheric effects. Furthermore, $\delta$D$_{wax}$ values in marine surface sediments along the Chilean coast (26–42° S) are consistent with $\delta$D$_{wax}$ values of soils and rivers from the adjacent continent, supporting the use of $\delta$D$_{wax}$ as a proxy for rainfall $\delta$D to study changes in paleohydrological conditions[42].

$\delta$D$_{wax}$ values reflect changes in rainfall amount, air temperature and moisture source. A D-depletion (more negative $\delta$D$_{wax}$ values) may result from increased rainfall amount (amount effect), lower condensation temperature (temperature effect), a more distal moisture source (rainout effect) and/or moisture from a colder source area (deuterium excess)[33,36,43]. In turn, less rainfall, higher temperature and/or a more proximal moisture source result in a D-enrichment (more positive $\delta$D$_{wax}$ values). At both 30° S and 41° S, SST and $\delta$D$_{wax}$ records are positively correlated (r = 0.8), which indicates a possible temperature effect on $\delta$D$_{wax}$ values. However, considering a temperature effect of up to 2‰/°C in the southern hemisphere mid-latitudes[44], temperature alone can explain at maximum half of long-term $\delta$D$_{wax}$ changes (Fig. 2a, b; Supplementary Tables S6, S7). Therefore, $\delta$D$_{wax}$ is considered here as reflecting variability in moisture source and rainfall amount in coastal Chile over the last ~50 ka.

### $\delta$D$_{wax}$ reflects changes in the moisture source during the Holocene
Continental and marine records suggest that at the beginning of the Holocene (12–8 ka) relatively dry conditions prevailed between 27° S and 43° S[16,40,45–54] (Fig. 3a), while wetter conditions were predominant south of 50° S[10,11]. This pattern is presumably related to a southward contraction of the SWW[10], which is in close agreement with relatively increased $\delta$D$_{wax}$ values at 30° S and 41° S compared to the deglaciation (Fig. 2a), suggesting a weaker influence of moisture from the high latitudes (Fig. 3b).

The middle Holocene (8–5 ka) was characterized by a further drying trend in south-central Chile[45,48,52,54,55], as well as south of 50° S[10,11]. This trend is reflected by a 6–7‰ increase in $\delta$D$_{wax}$ values at 41° S suggesting a weaker influence of moisture from the high latitudes, although part of the D-enrichment (1‰) may be related to the concurrent 1 °C SST warming (Fig. 2b). At 30° S, despite a slight decreasing trend (<2‰), which may be related to the <1 °C SST cooling, the $\delta$D$_{wax}$ values remained comparatively constant. A further poleward shift of the SWW combined with a decreasing wind strength may explain drier conditions in central and southern Chile during this time interval.

Humid conditions returned between 31° S and 51° S at the end of the Holocene (5–0 ka)[11,48,53,54,56,57], likely due to a northward expansion of the SWW[56] in line with a decrease in $\delta$D$_{wax}$ values at both 30° S and 41° S. The $\delta$D$_{wax}$ records off south-central Chile are thus reflecting changes in the high (D-depleted) versus low (D-enriched) latitude sources of moisture during the Holocene and confirm SWW latitudinal

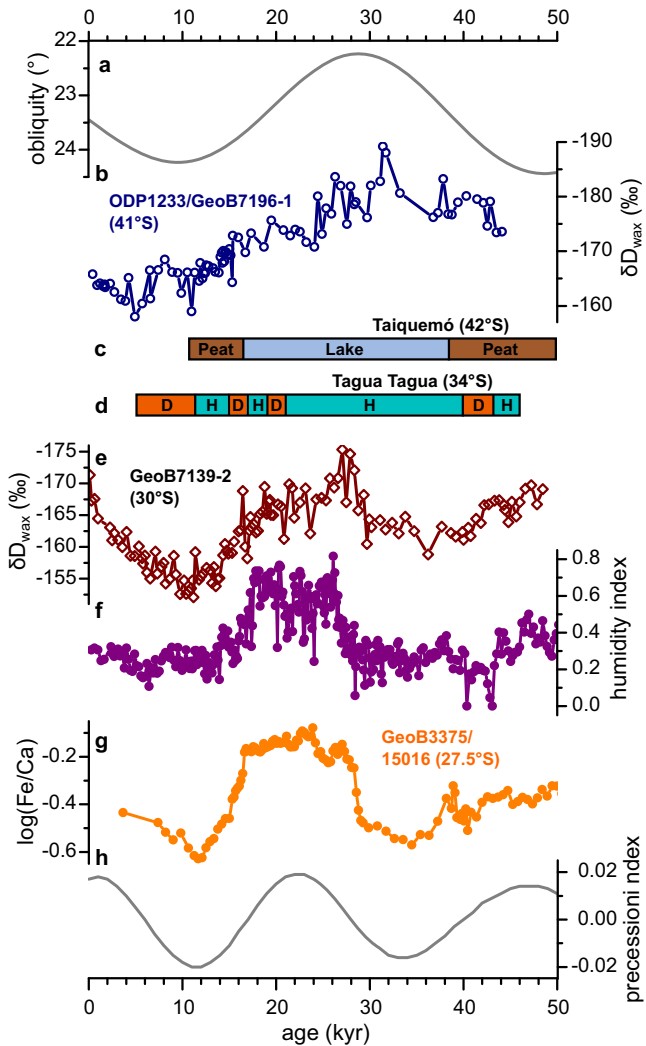

**Fig. 4 | Marine and continental moisture-related proxy records from coastal Chile over the last ~50 ka. a** Obliquity values[78]. **b** $\delta$D record of the $n$-$C_{31}$ alkane ($\delta D_{wax}$) off southern Chile (ODP1233/GeoB7196-1). **c** Synthesis of moisture-related records from the Taiquemó mire[58]. **d** Synthesis of moisture-related records from Laguna Tagua Tagua[40]. **e** $\delta$D record of the $n$-$C_{31}$ alkane ($\delta D_{wax}$) off central Chile (GeoB7139-2). **f** Humidity index record off central Chile (GeoB7139-2) based on sediment grain-size distributions (higher values indicate increased moisture). **g** Log(Fe/Ca) record off northern Chile (GeoB3375/15016)[18] indicating an increased relative amount of Fe-rich terrigenous sediment during humid conditions. **h** Precession index[78].

shifts. However, a minor effect of temperature on $\delta D_{wax}$ values cannot be completely ruled out.

## Implications for rainfall modulation in coastal Chile over the past 50 kyr

The SST records at 30° S and 41° S are positively correlated, as are the $\delta D_{wax}$ records. However, an $r$ value of 0.9 indicates that consistent SST variations occur along the Chilean margin, while a value of 0.6 suggests that moisture changes may have reacted to different forcing mechanisms at each latitude. Changes in $\delta D_{wax}$ values at 41° S are most likely related mainly to moisture source variability as the amount effect is restricted to lower, subtropical latitudes[43,44]. However, temperature may also affect $\delta D_{wax}$ variability to a minor extent as suggested from the Holocene part of the record. $\delta D_{wax}$ values are decreased around 34–26 ka and increased around 10–5 ka (Fig. 4b), indicating a continuously decreasing influence of subantarctic moisture between 30 and 10 ka. A moisture reconstruction on the adjacent land (42° S)

indicates hyper-humid last glacial conditions with maximum rainfall around 30 ka, which resulted in the transformation of a peatland into a lake[58] (Fig. 4c). This interval corresponds also to the maximum expansion of the Patagonian Ice Sheet and may be related to the obliquity minimum[59]. Indeed, shifts in SWW and associated oceanic fronts have been proposed to be related to orbital obliquity forcing[59–61] (Fig. 4a) in a way that phases of low (high) obliquity enhance (decrease) the meridional extra-tropical temperature gradient, increase (decrease) the atmospheric baroclinicity, and intensify (weaken) the SWW and associated storm tracks[62]. A concomitant Antarctic sea-ice extension further contributed to more equatorward located SWW[59]. Therefore, although the $\delta D_{wax}$ record is restricted to the last 45 kyr, it appears that changes in the moisture source at 41° S, and thus the location and strength of the SWW, are modulated by obliquity. Highest rainfall coupled with low temperatures favoured a maximal glacier extension in southern Chile around 30 ka[59].

At 30° S, we observe a different pattern. Decreased $\delta D_{wax}$ values occurred around 45–40 ka, 30–16 ka and after 5 ka (Fig. 4e). While decreased values around 28–25 ka, as well as in the late Holocene, may be related to an obliquity-modulated, high-latitude moisture source, yet this process cannot explain low $\delta D_{wax}$ values before 40 ka and during 25–18 ka at 30° S. Low $\delta D_{wax}$ values in subtropical Chile during these times might thus be related to higher rainfall amounts[20–22,24–26,63]. Indeed, two independent moisture-related proxy records confirm this assumption. Higher values of an index for continental moisture based on sediment grain-size distributions from the same sediment core GeoB7139-2 indicate increased moisture during 50–42 ka and 25–18 ka (Fig. 4f; Supplementary Table S8). A similar pattern is suggested by the Fe/Ca record of a sediment core located at 27.5° S, which estimates the proportion of Fe-rich terrigenous sediments relative to Ca-rich hemi-pelagic sediments[18] (Fig. 4g; Supplementary Table S9). It has been shown that this moisture record is related to a rainfall increase in north-central Chile during precession maxima. A stronger Pacific STJ leads to an increased moisture transport across the subtropical Pacific thus augmenting rainfall across subtropical and central Chile[18].

Therefore, the $\delta D_{wax}$ record at 30° S results from the combination of both the influence of subantarctic moisture around 30–25 ka and rainfall amount related to increased subtropical moisture around 50–42 ka and 25–18 ka. Similarly, a sedimentological, geochemical and palynological lake record of regional hydrologic balance suggests a humid phase between 40 and 10 ka at 34° S[40]. This record may thus also reflect the influence of a moisture maximum with a subantarctic source during the obliquity minimum and with a subtropical source during precession maxima (Fig. 4d). Strongly increased $\delta D_{wax}$ values at 30° S during the early Holocene most likely reflect the combination of both a reduced influence of moisture from the high latitudes related to the obliquity maximum and a lesser moisture input from subtropical latitudes during the precession minimum. Our results imply that the hydrological climate is strongly influenced by the subtropics at a precessional timescale at 30° S and by the subantarctic zone at an obliquity timescale at 41° S.

The subtropical moisture source variability as recorded at 30° S suggests a close link to atmosphere–ocean changes in the tropical Pacific at precessional timescales. Climate models point to austral winter zonal SST gradient changes across the equatorial Pacific as the key element controlling rainfall in subtropical Chile for these timescales[18]. Consistent with the $\delta D_{wax}$ record at 30° S, the proxy record at 27.5° S and two different climate models suggest increased rainfall in subtropical Chile when zonal SST gradients in the tropical Pacific were increased and the subtropical jet was stronger. These climate patterns related to precession changes resemble modern interannual variability during the El Niño Southern Oscillation (ENSO)[64,65]. The more humid interval during precession maxima also includes the LGM consistent with previous records suggesting La Niña-like states during this time interval[66]. In contrast to the modern climate

with enhanced rainfall in subtropical Chile during El Niño events, the paleo data at 27.5° S and 30° S indicate an opposite relation of long-term ENSO-like intervals, i.e., more moisture during La Niña-states. This finding suggests a variable response to ENSO forcing at different timescales and/or different overall climate background states, the last glacial and the Holocene. This is consistent with studies showing that ENSO activity was reduced during the last glacial[67,68], and therefore a different mean ENSO state might be expected. Most importantly, there is no consensus on ENSO intensity during the Holocene and the last glacial period, neither from proxy records nor model simulations[69]. Therefore, while ENSO accounts for about half of the winter rainfall variance in the modern climate[64,65], its influence on rainfall in central Chile was most likely reduced during the last glacial time and dominated by long-term ENSO-like variations of subtropical Chilean rainfall responding differently to SST changes in the tropical Pacific.

Reconstructing rainfall $\delta D$ in mid-latitudes represents a unique and promising approach to gain understanding into the spatio-temporal evolution of moisture sources (and rainfall amount when combined with other moisture-related proxies) to assess large-scale hydroclimatic changes. The present results imply that the hydrological patterns of the mid-latitudes are controlled by a combination of both the subtropics and the subantarctic zone at orbital timescales.

## Methods

### Sediment cores and age models

Sediment core GeoB7196-1 (41°0.00′ S, 74°26.99′ W) was retrieved 40 km off Chile at 851 m water depth during the PUCK expedition[70] on-board the German R/V Sonne. The age model of the core is based on 12 radiocarbon dates on mixed planktonic foraminifera samples (Supplementary Table S10). The radiocarbon dates were calibrated with the Marine20 calibration curve[71]. We assumed no regional deviation from the global reservoir effect off southern Chile during the Holocene[72]. At ODP1233 sediment cores have been drilled from four holes during ODP Leg 202 off southern Chile (41°0.01′ S, 74°26.99′ W; 40 km offshore; 838 m water depth) in a small basin on the upper continental slope[73]. The age model of the spliced sedimentary section covering the last 50 ka has been published in previous studies[74,75] and is mainly based on radiocarbon dates (Supplementary Table S11). Sediment core GeoB7139-2 was collected about 50 km offshore northern Chile (30°12′ S, 71°59′ W) at 3270 m water depth during the PUCK expedition[70] on-board the German R/V Sonne. The age model of the core has been published previously[16] (Supplementary Table S12). For the present study, ODP1233, GeoB7139-2, as well as GeoB3375/15016[18] age models were updated by recalibrating the radiocarbon dates with the Marine20 calibration curve[71] (Supplementary Table S13).

### n-Alkane extraction, purification and quantification

Compound-specific carbon and hydrogen isotope analyses of $n\text{-}C_{31}$ alkane were conducted on sediment samples from both ODP1233 ($n = 76$) and core GeoB7139-2 ($n = 123$). For core GeoB7196-1, the sediment samples ($n = 13$) were analysed for compound-specific hydrogen isotopes only and combined with ODP1233 data as both cores are from the same location. The sampling mean temporal resolution was 0.5 ka for ODP1233/GeoB7196-1 and 0.4 ka for core GeoB7139-2. The dried and finely ground sediment samples were extracted in a Thermo Scientific ASE200™ accelerated solvent extractor using a dichloromethane (DCM):methanol (MeOH) (9:1) solution at 1000 psi and 100 °C for three cycles lasting 5 min each. Known amounts of squalane were added prior to extraction as internal standard. Solvents were removed from total lipid extracts (TLEs) by rotary evaporation. Subsequently, residual water was removed over columns of $Na_2SO_4$ using hexane as eluent. TLEs were saponified in 0.5 ml of 0.1 M KOH in MeOH. Neutral fractions were obtained by liquid–liquid extraction using hexane after adding bi-distilled water.

The neutral fractions were separated over pipette columns of deactivated silica (1% $H_2O$) using hexane, DCM and DCM:MeOH (1:1) as eluents to yield hydrocarbon, ketone and polar fractions, respectively. Unsaturated compounds were removed from the hydrocarbon fractions by elution over $AgNO_3$-coated $SiO_2$ columns using hexane. $n$-Alkanes were quantified using a Thermo Scientific Focus gas chromatograph (GC) equipped with a 30 m Rxi™-5ms column (30 m × 0.25 mm × 0.25 µm) and a flame ionization detector. Quantification was achieved by comparing the peak areas to external standard solutions consisting of alkanes from $n\text{-}C_{16}$ to $n\text{-}C_{34}$. Repeated analyses of the standard solution indicate a quantification precision of <10%.

### Isotope analyses

$\delta D$ analyses of $n$-alkanes were conducted on a Thermo Scientific MAT 253™ Isotope Ratio Mass Spectrometer coupled via a GC IsoLink operated at 1420 °C to a Thermo Fisher Scientific TRACE GC equipped with a HP-5ms column (30 m × 0.25 mm × 1 µm). $\delta D$ values were calibrated against $H^2$ reference gas of known isotopic composition and are given in ‰ VSMOW. Each sample was measured in duplicate if sufficient material was available. Accuracy and precision were controlled by a lab internal $n$-alkane standard calibrated against the A4-Mix isotope standard (A. Schimmelmann, University of Indiana) every sixth measurements and by the daily determination of the $H_3^+$ factor. Measurement precision was determined by calculating the difference between the analysed values of each standard measurement and the long-term mean of standard measurements, which yielded a 1σ error of 3‰. $H_3^+$ factors varied between 5.0 and 5.2. Precision of the replicate analyses of the $n\text{-}C_{31}$ alkane was 1‰ on average. For samples which could only be analysed once and for the analytical replicates with better precision, the long-term precision of the standards (3‰) was assumed. $\delta^{13}C$ analyses of $n$-alkanes were conducted on a Thermo Fisher Scientific MAT 252 isotope ratio mass spectrometer coupled via a GC–C combustion interface with a nickel catalyser operated at 1000 °C to a Thermo Scientific Trace GC equipped with a HP-5ms column (30 m × 0.25 mm × 0.25 µm). Each sample was measured in duplicate. $\delta^{13}C$ values were calibrated against $CO_2$ reference gas of known isotopic composition and are given in ‰ VPDB. Accuracy and precision were determined by measuring $n$-alkane standards calibrated against the A4-Mix isotope standard every six measurement. Here the difference between the long-term means and the measured standard values yielded a 1σ error of 0.3‰. Accuracy and precision of the squalane internal standard were both 0.1‰. Precision of the replicate analyses of the $n\text{-}C_{31}$ alkane was 0.2‰ on average, respectively.

### Grain-size analysis and humidity index

The analysis of grain-size distributions of core GeoB7139-2 sediments was conducted on a Coulter laser granulometer (LS-200) over the size range 0.04–2000 µm. Prior to the grain-size analyses, bulk sediment subsamples were treated with 10% hydrochloric acid (HCl), 10% hydrogen peroxide ($H_2O_2$), and 1 N sodium hydroxide (NaOH) to remove carbonate, organic matter and biogenic opal components, respectively. For every sample, three consecutive grain-size measurements were run. An end-member modelling approach based on Weltje's end-member algorithm was used to determine the humidity index[15].

### Back trajectory analysis of rainy events along the Chilean coast

For each meteorological station (La Serena, Santo Domingo, Concepcion, Puerto Montt, Tortel) 50 independent rainy days (>5 mm/day) between 2006 and 2021 were identified and a back trajectory of the air parcels arriving at 500 m above ground level over the station was performed for 96 h. The HYSPLIT trajectory model[76] and the Global Data Assimilation System (GDAS) dataset (available at ftp://arlftp.arlhq.noaa.gov/pub/archives/gdas1) were used.

## Data availability

The source data are provided with this paper in the Supplementary Information file.

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

## Acknowledgements

J.K. was supported by the Leibniz Association. J.A.C. was supported by the Helmholtz Postdoc Programme (PD-001) and the Alfred Wegener Institute Helmholtz Centre for Polar and Marine Research, Bremerhaven, as for F.L. and N.R. E.S. was supported by the Cluster of Excellence 'The Ocean Floor—Earths Uncharted Interface'. R.D.P.-H. was partially supported by ANID/Fondecyt 1201810.

## Author contributions

J.K. and J.A.C. designed the study and analysed the data. J.K., E.S. and F.L. wrote the manuscript. J.A.C. and N.R. performed lipid extraction, purification and quantification. J.A.C. and E.S. performed the isotopic analyses. F.L. performed the XRF analysis. R.G. performed the back trajectory analysis. J.-B.S. performed the grain-size analysis. R.D.P.-H. performed the radiocarbon dating of core GeoB7169-1. J.K., J.A.C, E.S., F.L., N.R., R.G., J.-B.S. and R.D.P.-H. contributed to the discussion and interpretation.

## Funding

## Competing interests

The authors declare no competing interests.
