## [Peer Review File · Nature Communications]

Orbital modulation of subtropical versus subantarctic moisture sources in the southeast Pacific mid-latitudesREVIEWER COMMENTS

Reviewer #1 (Remarks to the Author):

This paper uses deuterium isotopes (δD) obtained from plant leaf waxes preserved in two marine sediment cores obtained off coastal Chile to reconstruct changes in rainfall amount and moisture sources over the last 45,000 years. The authors use these two cores to reconstruct a deuterium isotopic “gradient” that they then claim reflects changes in moisture amount and precipitation sources since MIS 3. The approach is pretty straightforward and, while not very new in terms of methodology, it is one of the first such leaf wax records from the southeastern Pacific Ocean and brings a large amount of valuable data to the table. The authors then use modeling results based on an ECEarth model (with no details provided regarding this model in the Methods section) to show that the most likely moisture source for rainfall events during precession maxima lies in the subtropics. According to the authors, the data presented here corroborate the findings proposed initially in another paper by this same research group (Lamy et al., 2019, DOI: 10.1073/pnas.1905847116) so in terms of the novelty implications of the results these are somewhat limited. One important question that immediately comes to mind which is not answered in either of these papers is why climates in south-central Chile are so much drier today compared to those during the LGM given that the SH is currently in a precession maximum.

The methods in this paper are more or less sound (although no description of the model used was provided) and at face value the results tend to support their conclusions. Although certainly useful, I do not fully agree with their interpretation of the δD “gradient”, especially during the Holocene. For example, while the δD data from the northern record show a clear negative trend after 10 ka, those from the southern record show an increasing trend, implying either a shift in moisture source or increased (decreased) evaporation in the latter (former) case. As their alkenone temperature data show cooling trends in both records during the Holocene, decreased evaporation could explain the northern δD trend in part but the southern trend would necessarily imply a shifting (i.e., subtropical) moisture source. So, does this imply a more southerly source for the northern record and a more subtropical source for the southern record? I do not see how this can work both ways even for the Holocene.

Finally, I do not understand why the authors did not attempt to correlate their record with continental reconstructions that span similar time intervals (e.g., Laguna de Tagua Tagua (LTT), <https://doi.org/10.1002/jqs.988>) which correlates with some of the wet phases described in their record (the 28-18 ka wet phase for example) but not others (18-10 ka, which is wet in the LTT record). To a similar degree, the same can be said for coastal records in central Chile which show very complex climate dynamics during the Mid-Late Holocene, see <https://doi.org/10.1002/jqs.2936>, <https://doi.org/10.1016/j.yqres.2006.04.003> for examples) which appear to be oversimplified in this leaf wax record.

In summary, the paper presents valuable data and is well-written for the most part (see minor comments below) but given that the major conclusions were already published in another paper almost four years ago by the same group, I do not see the novelty value in these new results beyond corroborating these earlier findings. Furthermore, the interpretation of the isotope data needs to be further fleshed out and compared to continental records from these latitudes. I agree with the authors that these records are few, but at least those that have already been published could provide valuable insights.

Minor comments

- L. 18: "Primordial" is an awkward word choice. Better to use "critical aspect".
- L. 34: "...and storm fronts can reach central Chile"
- L. 39: What do you mean by "quantitative" here? Most proxy records involve some sort of quantitative aspect (pollen counts, charcoal abundance, stable isotopes) so please clarify.
- L. 90: Given the lack of C4-dominated ecosystems in Chile this is to be expected. Although very small, a 1-2 per mil variation (probably due to water use efficiency, WUE) is clearly visible in these records and should be discussed. Especially because both records show similar patterns with higher values (implying increased WUE/aridity) during the Holocene.
- L. 95: "increased" instead of "enriched". Please remember that when discussing delta-notation these are unitless values that are not "depleted" nor "enriched" which would be an incorrect usage of these terms.
- L. 99: "... weaker isotopic gradients..."
- L. 122: I am not convinced that replacing $\Delta\delta D_{wax}$ for δD_{precip} is useful here. Why not just use one term throughout?
- L. 124: incorrect usage of "depleted", the correct term is "lower"
- L. 137: "moisture-related" instead of "humidity-related". Please replace all "humidity" with "moisture" throughout the text.
- L. 148: Where are the methods regarding the model used?
- L. 151: Many continental records from the Lake District are at odds with this interpretation.
- L. 164-167: Not sure I understand this last argument- based on your reconstruction we should be in a moisture maximum. Yet nothing appears to indicate that recent moisture levels reached those that occurred at the LGM. So, the reason for this is anthropogenic global warming and its impact on regional climate change?

Reviewer #2 (Remarks to the Author):

Overview: Kaiser et al present two new dD_{wax} records from sites along the Chilean margin - one at 30S and another at 40S - spanning the last 50,000 years. The records describe changes in hydroclimate at the glacial/interglacial and precessional time scales.

General Comments:

The new dD_{wax} records are welcome addition to paleoceanography along the Chilean margin, describing past isotopic changes in rainfall. However, the interpretation of the data needs to be expanded. The title focuses on the precession beat seen in the record from 30S, but the record at 41S is hardly discussed. I guess because this record does not follow precession - instead it shows more of a progressive change between the glacial and interglacial states. This highlights the importance of glacial conditions on modulating hydroclimate at 41S, but this isn't discussed in the text.

More generally the authors are selling the data short by interpreting the difference between the two sites rather than the two sites in isolation. The two sites are clearly controlled by different processes (30S, the STJ, but 40S, perhaps the eddy-driven jet, and things more directly related to the westerlies) and they show totally different evolution, so it doesn't make sense to me to take the difference between them. The authors do this I think only b/c they think temperature is overprinting the records, but I don't think that's a huge concern (see my detailed comment below). Why not discuss the full mix of subtropical and mid-latitude

processes at work here?

Final general comment is that, while I find these data interesting, I'm not sure what the high-profile advance is here that merits publication in Nature Communications. The authors aren't communicating in the text what the novel discovery is. The data are valuable but I think the paper would be better off in a speciality journal, where there would also be more space to discuss the differential climate dynamics affecting each site.

Specific Comments:

Line 18: "primordial" is not the right word here - 'fundamental', perhaps?

Line 34: "are reaching" should be "reach"

Line 37: this is the eddy-driven component of the jet, which I think is important to mention in more detail in the intro. The STJ does modify the storm tracks but ultimately the mid-latitude jet is bringing in the storms, hence rainfall variations in southern Chile are function of how both parts of the jet behave.

Line 45: not sure what "zonally asymmetrically" means here - describe what the authors found.

Line 49: direct proxy evidence of / Line 50: remains

Line 86: I agree that there is not much variation in $\delta^{13}\text{C}$ but Fig. S1 has a huge scale which I don't think is appropriate for the data. Zoom in between -36 and -30 so that the readers can assess whether or not there are any notable shifts.

Line 101: There is clearly no C4 signature in the $\delta^{13}\text{C}$ values, but that doesn't mean that vegetation change didn't influence δD . There is a lot of C3 grasses around in these ecosystems. Any sense of how much C3 grass coverage may or may not have affected these two sites? I know that ODP 1233 has a pollen record (Heusser et al., 2006); I'm not sure about the GeoB site, but it would be worth using any pollen information to check and see whether changing veg types could have had any influence or not.

Line 103: Is the amount effect really affecting rainfall isotopes at these locations? I think of the amount effect as mostly relevant in tropical locations and also in hyperarid places (re-evaporation of rainfall). Is there any evidence from station data (which I know is sparse in this region, but still worth checking) of an amount effect? If not, I think the moisture source argument is more compelling so I would work with that interpretation.

Line 112: First: correlation is denoted by r values not R^2 . R^2 is the coefficient of determination, a statistic for regressions, and is not the right statistic here since you are not fitting a model. Quote r values and show r values in Figure S2. Second: everything is correlated across the deglaciation, so just b/c temperature and δD are correlated doesn't mean that the relationship is causal. Temperature changes are expected to affect condensation temperatures at about 1-2 per mil per degree or so, and even that relationship can be pretty easily obscured by other more dominant processes in both subtropical and mid-latitude locations like the ones here. So, SST changes probably don't explain much of the deglacial/holocene trends and more likely, you are seeing some changes in moisture sources.

Line 114: By just jumping to discussing the difference b/t sites you are missing an opportunity to explore the mechanisms that controls dDp at each site, which are clearly different, given how different their time-evolution is.

Line 119 and the rest of the paragraph: "depleted at 30S/enriched at 41S": you mean the other way around, right? Site 1233 is much more depleted, whereas the GeoB site has some enrichment which seems precession-related. This whole paragraph seems to be mixed around since the high latitude site is more depleted during the glacial period.

Line 121: 30S is not in the ITCZ. As above, I don't think there is compelling evidence that amount effects are operating at these latitudes, unless you can show otherwise with station data.

Line 135: Fe/Ca - since it is a ratio the variations could be driven by Fe (terrigenous material) or CaCO₃ variations. Can you discern which one is in control by computing concentrations?

Line 157: Precessional changes are acting on much, much longer timescales than human timescales so it's specious to conflate the two as global warming is operating on decadal timescales. Delete.

Figure 4: I would compare the other proxy data to the dDwax record at the GeoB site only, since that's the site with the precessional pattern. ODP 1233 doesn't have much of a precession expression.

Figure 5: Plot higher precipitation as blue rather than red (which viewers associated with dry). Even better use a brown (dry) blue/green (wet) color bar.

Reviewer #3 (Remarks to the Author):

Kaiser et al. use paired leaf wax hydrogen isotope records from two marine sediment cores near the continental slope of the Chilean margin, to examine the changes in hydroclimate and specifically, moisture sources. This manuscript is neat and well-written, and I enjoy reading it. However, I do have some concerns about the $\Delta\delta D_{wax}$ and hope to see some explanation in the revision.

I feel the authors did a good job of explaining the factors that could potentially affect the leaf wax hydrogen isotopes in the past. Vegetation changes were minimal based on carbon isotope data (Fig. S1), and hydrogen isotopes are independent of SSTs from 43-18 kyr (Fig. S2). It is interesting to see the GeoB7139 records track the insolation very well (Fig. 3A and Fig. 4E), and it is certainly believable that it mostly tracks precipitation changes (Line 120). Meanwhile, hydrogen isotopes from ODP1233 look dislike insolation. I have difficulty fully understanding the actual meaning of $\Delta\delta D_{wax}$, without a clear estimated/assumed contribution of both. Then the authors present different inorganic proxies from 30°S or 27.5°S to infer hydroclimatic changes associated with the insolation as well, with a hidden assumption that they are more linked with rainfall amount, rather than moisture source. In my opinion, would it be easier, or more straightforward to measure such inorganic proxies (grain size and XRF) from ODP1233 and compare them with the δD_{wax} ?

For the model results in Fig. 5. Why JJA precipitation was used here? Did the authors look at

the precipitation from austral summer seasons? Also, I'm confused by the information delivered in Fig. 5. It seems the authors try to argue the moisture sources in central and southern Chile are mainly from subtropical regions. If this is true, does that imply that $\Delta\delta D_{wax}$ represents the rainfall difference between the two sites? Because the moisture source effect would be canceled out. It contrasts with the hypothesis that 30°S records are mostly rainfall amounts whereas the 41°S records are related to the moisture sources (Line120).

Minor suggestions:

Line 83, in this subsection, a few sentences are needed to introduce what the authors actually measured in this study. In the current format, I need to go to the Method section so I can know that the hydrogen and carbon isotopes of C31 alkanes are the proxies used in this study.

Line 408, Fig 1 map needs elements of lat and long.

REVIEWER COMMENTS

Reviewer #1 (Remarks to the Author):

This paper uses deuterium isotopes (δD) obtained from plant leaf waxes preserved in two marine sediment cores obtained off coastal Chile to reconstruct changes in rainfall amount and moisture sources over the last 45,000 years. The authors use these two cores to reconstruct a deuterium isotopic “gradient” that they then claim reflects changes in moisture amount and precipitation sources since MIS 3. The approach is pretty straightforward and, while not very new in terms of methodology, it is one of the first such leaf wax records from the southeastern Pacific Ocean and brings a large amount of valuable data to the table. The authors then use modeling results based on an ECEarth model (with no details provided regarding this model in the Methods section) to show that the most likely moisture source for rainfall events during precession maxima lies in the subtropics. According to the authors, the data presented here corroborate the findings proposed initially in another paper by this same research group (Lamy et al., 2019, DOI: 10.1073/pnas.1905847116) so in terms of the novelty implications of the results these are somewhat limited. One important question that immediately comes to mind which is not answered in either of these papers is why climates in south-central Chile are so much drier today compared to those during the LGM given that the SH is currently in a precession maximum.

We thank Reviewer #2 for appreciating the overall novelty of our leaf wax records in the South Pacific realm and are sorry that the reviewer concludes that the paleoclimatic implications are rather limited. The novelty of our paleoclimate implications resides in using δD as a proxy for changes in moisture sources, what was suggested by the model, but not shown empirically. This separation of moisture source has never been approached before and is critical for assessing large scale paleoclimate mechanism in the southern hemisphere. Our δD records suggest that during the LGM a combination of moisture from both the low (subtropical) and high (subantarctic) latitudes are consistent with northward shifted westerly winds and a stronger subtropical jet, what resulted in high rainfall in southern and central Chile. During the early Holocene however, the influence of the westerlies was decreased due to their obliquity-driven southward shift and the influence of subtropical humid air was decreased as well due to the precession minima. The modern climate in south-central Chile is so much drier today compared to the LGM most likely because of the relatively high obliquity resulting in a decreased influence of the westerlies. Note that we have now removed the model results as they have been published in Lamy et al. (2019). And we also suggest that precession mainly influence the δD signal at 30°S, but rather obliquity at 41°S.

The methods in this paper are more or less sound (although no description of the model used was provided) and at face value the results tend to support their conclusions. Although certainly useful, I do not fully agree with their interpretation of the δD “gradient”, especially during the Holocene. For example, while the δD data from the northern record show a clear negative trend after 10 ka, those from the southern record show an increasing trend, implying either a shift in moisture source or increased (decreased) evaporation in the latter (former) case. As their alkenone temperature data show cooling trends in both records during the Holocene, decreased evaporation could explain the northern δD trend in part but the southern trend would necessarily imply a shifting (i.e., subtropical) moisture source. So, does this imply a more southerly source for the northern record and a more subtropical source for the southern record? I do not see how this can work both ways even for the Holocene.

In agreement with the problematic interpretation of the δD gradient, it has been now removed from the manuscript. Concerning the Holocene part of the records, we have now analysed new samples from a sediment core (GeoB7196-1) from the same location as ODP Site 1233. Combined with ODP Site 1233 data, the δD record suggests now a slightly different pattern with a δD increase between 8 and 5 ka, which is related to a southward shift of the westerly winds and drier conditions as indicated by records from the adjacent land. Between 5-0 ka, δD was continuously decreasing due to a northward shift of the westerly winds and more humid conditions on the adjacent land. The Holocene

part of the δD records is now discussed in a new section of the discussion (please, see also below our answer to the next comment).

The model results have been removed from the manuscript as they have been published in Lamy et al. (2019). Therefore, a description of the model is not relevant anymore.

Finally, I do not understand why the authors did not attempt to correlate their record with continental reconstructions that span similar time intervals (e.g., Laguna de Tagua Tagua (LTT), <https://doi.org/10.1002/jqs.988>) which correlates with some of the wet phases described in their record (the 28-18 ka wet phase for example) but not others (18-10 ka, which is wet in the LTT record). To a similar degree, the same can be said for coastal records in central Chile which show very complex climate dynamics during the Mid-Late Holocene, see <https://doi.org/10.1002/jqs.2936>, <https://doi.org/10.1016/j.yqres.2006.04.003> for examples) which appear to be oversimplified in this leaf wax record.

In agreement with the reviewer's comment, the few existing, moisture-related proxy records from south-central Chile based on terrestrial archives have been added in synthesised forms in the figure (Figure 4) and are now discussed in context with our data. The Taiquemo mire record at 42 °S (Heusser et al., 1999) suggests hyper-humid conditions with maximum rainfall around 30 ka, what resulted in the transformation of the peatland into a lake. This is in close agreement with the δD values indicating an increased influence of subantarctic moisture and, thus, more humid conditions at 41 °S during the obliquity minimum centred around 30-28 ka. More humid conditions in southern Chile during the obliquity minimum have been suggested previously based on data of glacier advances in Patagonia (Fogwill et al., 2015), an aspect which is supported by the δD record at 41 °S. The records of the Laguna de Tagua Tagua located at 34 °S suggest humid conditions between 40-10 ka (Valero-Garcés et al., 2005) and may thus reflect the influence of both a moisture maximum with a subantarctic source during the obliquity minimum and a subtropical source during precession maxima.

Furthermore, we have now added new δD data for the Holocene part of the δD record at 41°S using a Holocene sediment core (GeoB7196-1) from the same location as ODP Site 1233. The combined δD record presents a pronounced (10‰) increase in δD values between 8 and 5 ka, followed by a constant decrease of ca. 7‰ towards the modern. This updated record together with the GeoB7139-2 δD record at 30 °S are now plotted in a new figure (Figure 3) dedicated to the Holocene part of the records (12-0 ka). δD variability during the Holocene is now discussed together with different land-based moisture reconstructions (Villagrán and Varela, 1990; Grosjean et al., 1997; Valero-Garcés et al., 1999; Jenny et al., 2002; Villa-Martínez et al., 2003; Moreno and León, 2003; Abarzúa et al., 2004; Moreno, 2004; Maldonado and Villagrán, 2006; Jara and Moreno, 2014; Frugone-Álvarez et al., 2017, 2020; Nehme et al., 2023) in a new section of the discussion. The main conclusion is that δD_p records off south-central Chile are reflecting changes in the high (D-depleted) versus low (D-enriched) latitude moisture sources during the Holocene and confirm latitudinal shifts of the westerly winds as main hydrological driver.

References:

*Abarzúa, A. M., Villagrán, C. & Moreno, P. I. Deglacial and postglacial climate history in east-central Isla Grande de Chiloé, southern Chile (43°S). *Quaternary Research* 62, 49–59 (2004).*

*Fogwill, C., Turney, C., Hutchinson, D., Taschetto, A. S. & England, M. H. Obliquity Control On Southern Hemisphere Climate During The Last Glacial. *Scientific Reports* 5, 11673 (2015).*

*Frugone-Álvarez, M., Latorre, C., Barreiro-Lostres, F., Giralt, S., Moreno, A., Polanco-Martínez, J., Maldonado, A., Carrevedo, M. L., Bernárdez, P., Prego, R., Delgado Huertas, A., Fuentealba, M. & Valero-Garcés, B. Volcanism and climate change as drivers in Holocene depositional dynamic of Laguna del Maule (Andes of central Chile – 36° S). *Climate of the Past* 16, 1097–1125 (2020).*

Frugone-Álvarez, M., Latorre, C., Giralt, S., Polanco-Martínez, J., Bernárdez, P., Oliva-Urcia, B., Maldonado, A., Carrevedo, M. L., Moreno, A., Delgado Huertas, A., Prego, R., Barreiro-Lostres, F. & Valero-Garcés, B. A 7000-year high-resolution lake sediment record from coastal central Chile (Lago Vichuquén, 34°S): implications for past sea level and environmental variability. *Journal of Quaternary Science* 32, 830–844 (2017).

Grosjean, M., Valero-Garcés, B. L., Geyh, M. A., Messerli, B., Schotterer, U., Schreier, H. & Kelts, K. Mid- and late Holocene limnogeology of Laguna del Negro Francisco, northern Chile, and its palaeoclimatic implications. *The Holocene* 7, 151–159 (1997).

Heusser, C. J., Heusser, L. E. & Lowell, T. V. Paleocology of the Southern Chilean Lake District-Isla Grande de Chiloé during Middle-Late Llanquihue Glaciation and Deglaciation. *Geografiska Annaler Series A, Physical Geography* 81, 231–84 (1999).

Jara, I. A. & Moreno, P. I. Climatic and disturbance influences on the temperate rainforests of northwestern Patagonia (40°S) since ~14,500 cal yr BP. *Quaternary Science Reviews* 90, 217–228 (2014).

Jenny, B., Valero-Garcés, B. L., Villa-Martínez, R., Urrutia, R., Geyh, M. & Veit, H. Early to Mid-Holocene Aridity in Central Chile and the Southern Westerlies: The Laguna Aculeo Record (34°S). *Quaternary Research* 58, 160–170 (2002).

Maldonado, A. & Villagrán, C. Climate variability over the last 9900 cal yr BP from a swamp forest pollen record along the semiarid coast of Chile. *Quaternary Research* 66, 246–258 (2006).

Moreno, P. I. & León, A. L. Abrupt vegetation changes during the last glacial to Holocene transition in mid-latitude South America. *Journal of Quaternary Science* 18, 787–800 (2003).

Moreno, P. I. Millennial-scale climate variability in northwest Patagonia over the last 15000 yr. *Journal of Quaternary Science* 19, 35–47 (2004).

Nehme, C., Todisco, D., Breitenbach, S. F. M., Couchoud, I., Marchegiano, M., Peral, M., Vonhof, H., Hellstrom, J., Tjallingii, R., Claeys, P., Borrero, L. & Martin, F. Holocene hydroclimate variability along the Southern Patagonian margin (Chile) reconstructed from Cueva Chica speleothems. *Global and Planetary Change* 222, 104050 (2023).

Valero-Garcés, B. L., Grosjean, M., Kelts, K., Schreier, H. & Messerli, B. Holocene lacustrine deposition in the Atacama Altiplano: facies models, climate and tectonic forcing. *Palaeogeography, Palaeoclimatology, Palaeoecology* 151, 101–125 (1999).

Valero-Garcés, B. L., Jenny, B., Rondanelli, M., Delgado-Huertas, A., Burns, S. J., Veit, H. & Moreno, A. Palaeohydrology of Laguna de Tagua Tagua (34°30'S) and moisture fluctuations in Central Chile for the last 46000 yr. *Journal of Quaternary Science* 20, 625–641 (2005).

Villagrán, C. & Varela, J. Palynological evidence for increased aridity on the central Chilean coast during the Holocene. *Quaternary Research* 34, 198–207 (1990).

Villa-Martínez, R., Villagrán, C. & Jenny, B. The last 7500 cal yr B.P. of westerly rainfall in Central Chile inferred from a high resolution pollen record from Laguna Aculeo (34°S). *Quaternary Research* 60, 284–293 (2003).

In summary, the paper presents valuable data and is well-written for the most part (see minor comments below) but given that the major conclusions were already published in another paper almost four years ago by the same group, I do not see the novelty value in these new results beyond corroborating these earlier findings. Furthermore, the interpretation of the isotope data needs to be further fleshed out and compared to continental records from these latitudes. I agree with the authors

that these records are few, but at least those that have already been published could provide valuable insights.

We agree with Reviewer #1 that our δD records provide novel information of rainfall amount and changes in moisture sources. This geochemical information, which cannot be derived from traditional rainfall proxies, provides direct proxy evidence for the previously published model indication for precessional cycles in subtropical moisture supply. This is confirmed by our northern record at 30°S. Furthermore, our revised interpretation of the δD record at 41°S presents a novel aspect by suggesting an obliquity modulated variability in the high latitude (subantarctic) moisture source at 41°S, while precession modulates changes in the low-latitude (subtropical) moisture source. This dual aspect is novel and provides new insights into paleoclimatic interactions of subtropical and subantarctic climates and their modulation through the Southern Hemisphere westerly winds. The δD records are now interpreted for each site individually and compared to available continental proxy records. Furthermore, a focus on the Holocene part of the records has been added in order to test and validate the δD moisture proxy against continental records.

Minor comments

L. 18: “Primordial” is an awkward word choice. Better to use “critical aspect”.

We agree and have edited the text accordingly (line 20 of the revised version of the manuscript).

L. 34: “...and storm fronts can reach central Chile”

We changed the wording into “... and occasionally extra-tropical cyclones reach central Chile” (line 38).

L. 39: What do you mean by “quantitative” here? Most proxy records involve some sort of quantitative aspect (pollen counts, charcoal abundance, stable isotopes) so please clarify.

In agreement with the suggestion by Reviewer #2, this sentence has been changed into: “Direct proxy evidence of SWW variability in the past remains difficult to obtain.” (line 44)

L. 90: Given the lack of C4-dominated ecosystems in Chile this is to be expected. Although very small, a 1-2 per mil variation (probably due to water use efficiency, WUE) is clearly visible in these records and should be discussed. Especially because both records show similar patterns with higher values (implying increased WUE/aridity) during the Holocene.

We thank Reviewer #1 for pointing us to this potential impact. While the effect of aridity and water use efficiency is probably almost absent in the humid climate at 41 °S, these parameters may have indeed influenced the δD values at 30 °S. However, considering the uncertainty of the isotopic analyses ($\pm 3\%$), changes of 1-2‰ are most likely not significant. Furthermore, a recent study based on δD analyses on Chilean soils, rivers and marine sediments between 26-42 °S suggests that the effect of aridity on δD values is significant only in the hyperarid zone located north of 28 °S (Gaviria-Lugo et al., 2023; now cited in the new version of the manuscript). Therefore, the eventual effect of water use efficiency/aridity on the δD records was not discussed further.

Reference:

*Gaviria-Lugo, N., Lauchli, C., Wittmann, H., Bernhardt, A., Frings, P., Mohtadi, M., Rach, O. & Sachse, D. Climatic controls on leaf wax hydrogen isotope ratios in terrestrial and marine sediments along a hyperarid-to-humid gradient. *Biogeosciences* 20, 4433–4453 (2023).*

L. 95: “increased” instead of “enriched”. Please remember that when discussing delta-notation these are unitless values that are not “depleted” nor “enriched” which would be an incorrect usage of these terms.

We agree and have replaced “enriched” and “depleted” by “increased” and “decreased”, respectively, throughout the manuscript.

L. 99: “... weaker isotopic gradients...”

This line has been deleted as the δD gradient aspect has been removed from the manuscript.

L. 122: I am not convinced that replacing $\Delta\delta D_{wax}$ for δD_{precip} is useful here. Why not just use one term throughout?

We understand the reviewer’s concern. We added now: “Therefore, δD_{wax} is considered here as reflecting mainly variability in moisture source and rainfall amount (δD_p) in coastal Chile over the last ~50 ka.” (lines 122–124). Beyond this sentence defining δD_p , only δD_p has been used in the text.

L. 124: incorrect usage of “depleted”, the correct term is “lower”

Thank you, it has been corrected.

L. 137: “moisture-related” instead of “humidity-related”. Please replace all “humidity” with “moisture” throughout the text.

We agree and we have changed this accordingly, except for the index defined as “humidity index”.

L. 148: Where are the methods regarding the model used?

The modelling results have been removed from the manuscript as they have been published in Lamy et al. (2019) and are less relevant for the modified version of the manuscript.

L. 151: Many continental records from the Lake District are at odds with this interpretation.

To clarify our interpretations, we have substantially rewritten and extended this part of the manuscript (lines 149–152). The main moisture source at 41°S is now interpreted as originating from the subantarctic zone and responding to obliquity changes in agreement with records from Isla Grande de Chiloé (Heusser et al., 1999) and advances of glaciers from the Patagonian ice sheet (Fogwill et al., 2015).

References:

Fogwill, C., Turney, C., Hutchinson, D., Taschetto, A. S. & England, M. H. Obliquity Control On Southern Hemisphere Climate During The Last Glacial. Scientific Reports 5, 11673 (2015).

Heusser, C. J., Heusser, L. E. & Lowell, T. V. Paleoecology of the Southern Chilean Lake District-Isla Grande de Chiloé during Middle-Late Llanquihue Glaciation and Deglaciation. Geografiska Annaler Series A, Physical Geography 81, 231–84 (1999).

L. 164-167: Not sure I understand this last argument- based on your reconstruction we should be in a moisture maximum. Yet nothing appears to indicate that recent moisture levels reached those that occurred at the LGM. So, the reason for this is anthropogenic global warming and its impact on regional climate change?

We thank the reviewer for pointing us to this important argument, which has been removed in the modified version of the manuscript. We now suggest that the hydrological patterns of the mid-latitudes are controlled by forcing factors related to both the subtropics and the subantarctic zone at orbital timescales. The combination of a high obliquity, reducing the influence of the westerly winds, and a low precession index, reducing the influence of subtropical moisture, may explain the arid conditions during the early Holocene.

Reviewer #2 (Remarks to the Author):

Overview: Kaiser et al present two new δD_{wax} records from sites along the Chilean margin - one at 30S and another at 40S - spanning the last 50,000 years. The records describe changes in hydroclimate at the glacial/interglacial and precessional time scales.

General Comments:

The new δD_{wax} records are welcome addition to paleoceanography along the Chilean margin, describing past isotopic changes in rainfall. However, the interpretation of the data needs to be expanded. The title focuses on the precession beat seen in the record from 30S, but the record at 41S is hardly discussed. I guess because this record does not follow precession - instead it shows more of a progressive change between the glacial and interglacial states. This highlights the importance of glacial conditions on modulating hydroclimate at 41S, but this isn't discussed in the text.

More generally the authors are selling the data short by interpreting the difference between the two sites rather than the two sites in isolation. The two sites are clearly controlled by different processes (30S, the STJ, but 40S, perhaps the eddy-driven jet, and things more directly related to the westerlies) and they show totally different evolution, so it doesn't make sense to me to take the difference between them. The authors do this I think only b/c they think temperature is overprinting the records, but I don't think that's a huge concern (see my detailed comment below). Why not discuss the full mix of subtropical and mid-latitude processes at work here?

We agree with this important suggestion of Reviewer #2 and, therefore, we have removed the δD gradient record and we are now interpreting the two sites individually. While precession modulates changes in the low-latitude (subtropical) moisture source, as we have argued in the previous version of the manuscript, we now suggest an obliquity modulated variability of the higher latitude (subantarctic) moisture source at 41°S, i.e. the Southern Hemisphere westerly winds. In the revised manuscript, the discussion is now focusing on the interplay between subtropical and subantarctic moisture sources in south-central Chile during the last ~50 ka.

Final general comment is that, while I find these data interesting, I'm not sure what the high-profile advance is here that merits publication in Nature Communications. The authors aren't communicating in the text what the novel discovery is. The data are valuable but I think the paper would be better off in a speciality journal, where there would also be more space to discuss the differential climate dynamics affecting each site.

The novelty resides in using δD as a proxy for changes in moisture source, what cannot be done by traditional rainfall proxies. The δD records are now interpreted for each site individually. Our revised interpretation of the δD record at 41°S suggests an obliquity modulated variability in the high latitude (subantarctic) moisture source at 41°S, while precession modulates changes in the low-latitude (subtropical) moisture source. This dual aspect is novel and provides new insights into paleoclimatic interactions of subtropical and subantarctic climates and their modulation through the Southern Hemisphere westerly winds. Therefore, we do think that our study has important implications, which are worth to be published in Nature Communications.

Specific Comments:

Line 18: "primordial" is not the right word here - 'fundamental', perhaps?

We agree with the reviewer and changed "primordial" to "a critical aspect" (line 20 of the revised version of the manuscript).

Line 34: "are reaching" should be "reach"

Thank you, it has been changed accordingly (line 38).

Line 37: this is the eddy-driven component of the jet, which I think is important to mention in more detail in the intro. The STJ does modify the storm tracks but ultimately the mid-latitude jet is bringing in the storms, hence rainfall variations in southern Chile are function of how both parts of the jet behave.

We agree with Reviewer #2 that a more detailed introduction of the jets and resulting rainfall variations in Chile would strengthen our manuscript and the interpretation of our paleo-data. In the revised manuscript, we therefore added more information, especially regarding the interannual variability of Split Jet following Chiang et al. (2014) (cited in the main text). The reviewer is correctly stating that the STJ primarily affects the location of storm tracks but the storms themselves originate from the mid-latitude jet. We revised the introductory text accordingly (lines 32–43).

Reference:

Chiang, J.C.H, Lee, S.-Y., Putnam, A.E. & Wang, X. South Pacific Split Jet, ITCZ shifts, and atmospheric North–South linkages during abrupt climate changes of the last glacial period. Earth and Planetary Science Letters 406, 233–246 (2014).

Line 45: not sure what "zonally asymmetrically" means here - describe what the authors found.

We agree that “zonally asymmetric” might need some further explanation. It refers to the overall observation that the austral winter jet stream configuration in the South Pacific is different from the Atlantic and Indian Ocean sectors and thus the westerly winds as whole are not zonally symmetric around the globe. The Pacific sector is characterized by a split of the high-altitude jet stream into strong sub-tropical and subpolar jets, and a weaker mid-latitude jet.

We reworded this section in the main text (lines 53–57): “A one-million-year rainfall record based on continental slope sediments off the southern Atacama Desert provides evidence that the austral winter South Pacific SWW configuration varied zonally asymmetric on precessional timescales (19/23-kyr cycles), namely stronger SWW in the South Pacific sector and weaker in the other ocean basins during precession maxima¹⁸.”

Line 49: direct proxy evidence of / Line 50: remains

Thanks you, we changed this accordingly (line 44).

Line 86: I agree that there is not much variation in $\delta^{13}C$ but Fig. S1 has a huge scale which I don't think is appropriate for the data. Zoom in between -36 and -30 so that the readers can assess whether or not there are any notable shifts.

We agree and restricted to -16 to -35‰ and an axis break between -24 and -30‰ has been added. The aim of the figure remains to show that there was no significant change between C_3 and C_4 plants that could affect significantly the δD values of the records.

Line 101: There is clearly no C4 signature in the $\delta^{13}\text{C}$ values, but that doesn't mean that vegetation change didn't influence δD . There is a lot of C3 grasses around in these ecosystems. Any sense of how much C3 grass coverage may or may not have affected these two sites? I know that ODP 1233 has a pollen record (Heusser et al., 2006); I'm not sure about the GeoB site, but it would be worth using any pollen information to check and see whether changing veg types could have had any influence or not.

Unfortunately, there are no pollen data for core GeoB7139-2. Pollen records from the adjacent land (Laguna Tagua Tagua - 34°S; Heusser, 1990; Valero-Garcés et al., 2005) suggest as regional vegetation a beech woodland with podocarps before ca. 40 ka, a beech-podocarp woodland between ca. 40–28 ka, and a Podocarpaceae and Fagaceae forest and extended wetland areas between ca. 28–14 ka. Thus, until 14 ka the record is dominated by arboreal (Nothofagus) and woodland (Prumnopitys) pollen. Between 14–6 ka, the forest opened and large areas were colonised by grasses and shrubs (Chenopodiaceae, Cyperaceae, Asteraceae, Typha), indicating drier conditions. After 6 ka, the regional forest of Podocarpaceae and Fagaceae dominated again. Therefore, only the increase in grasses between 14–6 ka may have influenced δD values. However, pollen derived from monocotyledonous plants such as Cyperaceae and Typha remained low (<20%) on the one hand. On the other hand, as n-alkanes derived from monocotyledonous plants are D-depleted (Liu et al., 2016), a decrease in δD values would be expected, but the δD record increased during this time interval. Therefore, an effect of vegetation change on the δD record at 30°S is most likely not relevant.

The ODP Site 1233 record at 41°S shows that arboreal pollen (Nothofagus dombeyi) were around 60% between 50–18 ka, and decreased thereafter to ca. 40% (Heusser et al., 2006). The pollen of graminoids were relatively stable around 20% between 50–0 ka. Myrtaceae pollen are in relatively high amount (up to 25%) between ca. 18–10 ky. However, Myrtaceae are dicotyledonous plants and therefore no effect on δD values are expected. Fern (Filicinae) pollen are increasing from 10% to 30% between 18–0 ka. Although assessments of δD values from ferns are limited, studies have shown that fern δD values are similar to woody plants (He et al., 2020) and suggested that relative contributions of leaf waxes from ferns do not have predictable effects on sedimentary δD values (Ladd et al., 2021).

Furthermore, a recent study based on δD analyses on Chilean soils, rivers and marine sediments shows that between 28–42 °S δD values reflect rain δD values in the modern settings and are, therefore, not biased by changes in the type of vegetation along coastal Chile (Gaviria-Lugo et al., 2023).

To conclude, changing vegetation types did most likely not have significant influence on the δD records at both 30°S and 41°S. As the pollen records support the $\delta^{13}\text{C}$ results, they were not discussed in the manuscript.

References:

Gaviria-Lugo, N., Lauchli, C., Wittmann, H., Bernhardt, A., Frings, P., Mohtadi, M., Rach, O. & Sachse, D. Climatic controls on leaf wax hydrogen isotope ratios in terrestrial and marine sediments along a hyperarid-to-humid gradient. Biogeosciences 20, 4433–4453 (2023).

Heusser, C. J. Ice age vegetation and climate of subtropical Chile. Palaeogeography, Palaeoclimatology, Palaeoecology 80, 107–127 (1990).

Heusser, L., Heusser, C. J. & Pisias, N. Vegetation and climate dynamics of southern Chile during the past 50,000 years: results of ODP Site 1233 pollen analysis. Quaternary Science Reviews 25, 474–485 (2006).

Ladd, S. N., Maloney, A. E., Nelson, D. B., Prebble, M., Camperio, G., Sear, D. A., Hassall, J. D., Langdon, P. G., Sachs, J. P. & Dubois, N. Leaf wax hydrogen isotopes as a hydroclimate proxy in the tropical Pacific. Journal of Geophysical Research: Biogeosciences 126, e2020JG005891 (2021).

Liu, J., Liu, W., An, Z. & Jang, H. Different hydrogen isotope fractionations during lipid formation in higher plants: Implications for paleohydrology reconstruction at a global scale. *Scientific Reports* 6, 19711 (2016).

Valero-Garcés, B. L., Jenny, B., Rondanelli, M., Delgado-Huertas, A., Burns, S. J., Veit, H. & Moreno, A. Palaeohydrology of Laguna de Tagua Tagua (34°30'S) and moisture fluctuations in Central Chile for the last 46000 yr. *Journal of Quaternary Science* 20, 625–641 (2005).

Line 103: Is the amount effect really affecting rainfall isotopes at these locations? I think of the amount effect as mostly relevant in tropical locations and also in hyperarid places (re-evaporation of rainfall). Is there any evidence from station data (which I know is sparse in this region, but still worth checking) of an amount effect? If not, I think the moisture source argument is more compelling so I would work with that interpretation.

We understand the reviewer's concern. Firstly, an amount effect was suggested only for the δD record at 30°S, and not at 41°S. Then, based on available seasonal station data for the period 1991–2015 at La Serena (30°S; see table below), there is indeed a decrease in rain δD in winter when rainfall amount (Pmm) is highest, but decreased rainfall deuterium excess (d-ex) as well as air temperature (T °C) can also partly explain the δD decrease. As all physical atmospheric isotope effects are acting regardless of location and time, the only difference is which effect is dominant at a specific location and a specific time.

La Serena (1991-2015)					
Months	d18O	d2H	d-ex	Pmm	T (°C)
SON (spring)	-4.6	-23.7	13.3	11.4	14.0
DJF (summer)	-	-	-	-	-
MAM (autumn)	-4.5	-26.7	8.9	26.6	15.0
JJA (winter)	-5.2	-32.6	8.8	68.5	12.0

Based on our data, decreased δD values at 30°S during 26–16 ka can be explained only by the amount effect as the influence of subantarctic moisture is decreasing during this period as indicated by increasing δD values at 41°S. Therefore, we suggest that decreased δD values at 30°S during the late glacial are related to increased rainfall as implied by the other moisture-related proxy (humidity index). Except for this time interval at 30°S, the δD records are interpreted as related to variability in moisture source.

Line 112: First: correlation is denoted by r values not R². R² is the coefficient of determination, a statistic for regressions, and is not the right statistic here since you are not fitting a model. Quote r values and show r values in Figure S2. Second: everything is correlated across the deglaciation, so just b/c temperature and dD are correlated doesn't mean that the relationship is causal. Temperature changes are expected to affect condensation temperatures at about 1-2 per mil per degree or so, and even that relationship can be pretty easily obscured by other more dominant processes in both subtropical and mid-latitude locations like the ones here. So, SST changes probably don't explain much of the deglacial/holocene trends and more likely, you are seeing some changes in moisture sources.

We agree that this issue is complicated and we have now addressed this in different way. We have removed Figure S2, and the relationship between δD and temperature is discussed with regard to a possible minor effect of temperature on δD variability: “At both 30°S and 41°S, sea surface temperature (SST) and δD_{wax} records are positively correlated ($r = 0.8$; $p < 0.01$), what indicates a possible temperature effect on δD_{wax} values. However, considering a temperature effect of up to 2‰/°C in the southern hemisphere mid-latitudes⁴⁰, temperature alone can explain at maximum half of long-term δD_{wax} changes (Fig. 2A-B). Therefore, δD_{wax} is considered here as reflecting mainly variability in moisture source and rainfall amount (δD_p) in coastal Chile over the last ~50 ka.” (lines 119–124).

Line 114: By just jumping to discussing the difference b/t sites you are missing an opportunity to explore the mechanisms that controls δD_p at each site, which are clearly different, given how different their time-evolution is.

We thank the reviewer for pointing us to this important approach. We have now removed the δD gradient aspect of the manuscript and we are discussing the δD records individually. By following this approach, we suggest different control mechanisms at 30°S and 41°S. The δD record at 30°S is modulated by precession as suggested in the submitted version. However, we propose now that obliquity modulated the δD record at 41°S in a way that phases of low (high) obliquity enhance (decrease) the meridional extratropical temperature gradient, increase (decrease) the atmospheric baroclinicity, and intensify (weaken) the southern hemisphere westerly winds and associated storm tracks as main moisture source. Therefore, decreased δD values at 41°S between 34–26 ka reflect an increased influence of subantarctic moisture. This is in line with increased rainfall (Heusser et al., 1999) and glacier extension (Fogwill et al., 2015) on land. An obliquity modulated moisture source in the southern hemisphere mid-latitudes has not been evidenced so far (to the best of our knowledge). This aspect is thoroughly discussed in “Implications for long-term rainfall modulation in coastal Chile over the past ~50 kyr” (lines 142–189).

References:

Fogwill, C., Turney, C., Hutchinson, D., Taschetto, A. S. & England, M. H. Obliquity Control On Southern Hemisphere Climate During The Last Glacial. Scientific Reports 5, 11673 (2015).

Heusser, C. J., Heusser, L. E. & Lowell, T. V. Paleoecology of the Southern Chilean Lake District-Isla Grande de Chiloé during Middle-Late Llanquihue Glaciation and Deglaciation. Geografiska Annaler Series A, Physical Geography 81, 231–84 (1999).

Line 119 and the rest of the paragraph: "depleted at 30S/enriched at 41S": you mean the other way around, right? Site 1233 is much more depleted, whereas the GeoB site has some enrichment which seems precession-related. This whole paragraph seems to be mixed around since the high latitude site is more depleted during the glacial period.

This is correct. We have now reworded this paragraph and moved it into the part of the discussion about the implications for long-term rainfall modulation (lines 142–189).

Line 121: 30S is not in the ITCZ. As above, I don't think there is compelling evidence that amount effects are operating at these latitudes, unless you can show otherwise with station data.

We thank Reviewer #2 for critically reviewing the amount effect and we now address this issue in the revised discussion. As mentioned above, available seasonal station data for the period 1991-2015 at La Serena (30°S) indicate a decrease in rain δD in winter when rainfall amount (Pmm) is highest, but decreased rainfall deuterium excess (d-ex) as well as air temperature (T °C) can also partly explain the δD decrease in the modern settings. As the temperature increase between 26–16 ka cannot explain decreased δD values at 30°S, these latter are related most likely to increased rainfall as implied by the other moisture-related proxies.

Line 135: Fe/Ca - since it is a ratio the variations could be driven by Fe (terrigenous material) or CaCO₃ variations. Can you discern which one is in control by computing concentrations?

As illustrated below, the Fe/Ca record of core GeoB7139-2 is controlled by both Fe and Ca. However, Ca is more strongly correlated to Fe/Ca (log ratio), what suggests that the ratio is possibly not mirroring terrigenous inputs only. As the humidity index record of core GeoB7139-2 has already been partly published (Bernhardt et al., 2017) and has been validated as relevant moisture proxy in different settings (Stuut and Lamy, 2004), the Fe/Ca record of core GeoB7139-2 has been removed from the manuscript, and only the humidity index record is now shown in Figure 4.

Concerning the Fe/Ca record from core GeoB3375/15016 published in Lamy et al. (2019), it has been shown that it is controlled by terrigenous input (Lamy et al., 2019 – supplementary information).

Reference:

Bernhardt, A., Schwanghart, W., Hebbeln, D., Stuut, J.-B. W. & Strecker, M. R. Immediate propagation of deglacial environmental change to deep-marine turbidite systems along the Chile convergent margin. *Earth and Planetary Science Letters* 473, 190–204 (2017).

Lamy, F., Chiang, J. C. H., Martínez-Méndez, G., Thierens, M., Arz, H. W., Bosmans, J., Hebbeln, D., Lambert, F., Lembke-Jene, L. & Stuut, J.-B. W. Precession modulation of the South Pacific westerly wind belt over the past million years. *Proceedings of the National Academy of Sciences of the United States of America* 116, 23455–23460 (2019).

Stuut, J. B. W. & Lamy, F. Climate variability at the southern boundaries of the Namib (southwestern Africa) and Atacama (northern Chile) coastal deserts during the last 120,000 yr. *Quaternary Research* 62, 301–309 (2004).

Line 157: Precessional changes are acting on much, much longer timescales than human timescales so it's specious to conflate the two as global warming is operating on decadal timescales. Delete.

We agree. This part has been deleted.

Figure 4: I would compare the other proxy data to the dDwax record at the GeoB site only, since that's the site with the precessional pattern. ODP 1233 doesn't have much of a precession expression.

We agree with Reviewer #1 and this figure has been substantially changed in the revised version.

Figure 5: Plot higher precipitation as blue rather than red (which viewers associated with dry). Even better use a brown (dry) blue/green (wet) color bar.

This figure has been removed from the manuscript.

Reviewer #3 (Remarks to the Author):

Kaiser et al. use paired leaf wax hydrogen isotope records from two marine sediment cores near the continental slope of the Chilean margin, to examine the changes in hydroclimate and specifically, moisture sources. This manuscript is neat and well-written, and I enjoy reading it. However, I do have some concerns about the $\Delta\delta D_{wax}$ and hope to see some explanation in the revision.

I feel the authors did a good job of explaining the factors that could potentially affect the leaf wax hydrogen isotopes in the past. Vegetation changes were minimal based on carbon isotope data (Fig. S1), and hydrogen isotopes are independent of SSTs from 43-18 kyr (Fig. S2). It is interesting to see the GeoB7139 records track the insolation very well (Fig. 3A and Fig. 4E), and it is certainly believable that it mostly tracks precipitation changes (Line 120). Meanwhile, hydrogen isotopes from ODP1233 look dislike insolation. I have difficulty fully understanding the actually meaning of $\Delta\delta D_{wax}$, without a clear estimated/assumed contribution of both. Then the authors present different inorganic proxies from 30°S or 27.5°S to infer hydroclimatic changes associated with the insolation as well, with a hidden assumption that they are more linked with rainfall amount, rather than moisture source. In my opinion, would it be easier, or more straightforward to measure such inorganic proxies (grain size and XRF) from ODP1233 and compare them with the δD_{wax} ?

We thank Reviewer #3 for this important suggestion. We have now removed the δD gradient aspect of the manuscript and reinterpreted the δD record from ODP Site 1233. Each site is now discussed individually. The main conclusions are that while moisture source is controlled by precession at 30°S, as suggested in the submitted version, it is controlled by obliquity at 41°S. Therefore, the hydrological patterns of the mid-latitudes are controlled by a combination of both the subtropics and the subantarctic zone at orbital timescales.

Concerning inorganic proxy records from ODP Site 1233, no grain size data are available as far as we know. The Fe/Ca record of ODP Site 1233 cannot be used as direct rainfall proxy as Fe is controlled by both glacier and rainfall variability at 41°S (Lamy et al., 2002, 2004). However, we have now added a moisture-related record from the land adjacent to ODP Site 1233 (Tiquemó mire - 42°S; Heusser et al., 1999). This record shows that the peatland turned into a lake between 38–18 ka, what indicates increased moisture during this period, in close agreement with an increased subantarctic moisture source (i.e. an increased influence of the westerly winds) at 41°S as suggested by decreased δD values during this time.

References:

Heusser, C. J., Heusser, L. E. & Lowell, T. V. Paleoecology of the Southern Chilean Lake District-Isla Grande de Chiloé during Middle-Late Llanquihue Glaciation and Deglaciation. Geografiska Annaler Series A, Physical Geography 81, 231–84 (1999).

Lamy, F., Kaiser, J., Ninnemann, U., Hebbeln, D., Arz, H. W. & Stoner, J. Antarctic Timing of Surface Water Changes off Chile and Patagonian Ice Sheet Response. Science 304, 1959–1962 (2004).

Lamy, F., Rühlemann, C., Hebbeln, D. & Wefer, G. High- and low-latitude climate control on the position of the southern Peru-Chile Current during the Holocene. Paleoceanography 17, 2001PA000727 (2002).

For the model results in Fig. 5. Why JJA precipitation was used here? Did the authors look at the precipitation from austral summer seasons? Also, I'm confused by the information delivered in Fig. 5. It seems the authors try to argue the moisture sources in central and southern Chile are mainly from subtropical regions. If this is true, does that imply that $\Delta\delta D_{wax}$ represents the rainfall difference between the two sites? Because the moisture source effect would be cancelled out. It contrasts with the hypothesis that 30°S records are mostly rainfall amounts whereas the 41°S records are related to the moisture sources (Line 120).

We removed Figure 5 as all details are discussed in the previous publication of Lamy et al. (2019). As well, the δD gradient aspect has been removed as both records are now discussed individually. We suggest now that the δD record at 30°S is modulated by precession, while the δD record at 41°S by obliquity. These two aspects are now thoroughly discussed in the part of the discussion about the implications for long-term rainfall modulation in coastal Chile over the past ~50 kyr (lines 142–189).

Minor suggestions:

Line 83, in this subsection, a few sentences are needed to introduce what the authors actually measured in this study. In the current format, I need to go to the Method section so I can know that the hydrogen and carbon isotopes of C₃₁ alkanes are the proxies used in this study.

We agree and we have now clarified earlier in the text which analyses have been done in our study. The analyses done are mentioned at the end of the Introduction, in the Results, and in the Methods. We have modified the sentence in the Results into: “Leaf wax deuterium (δD_{wax}) and carbon ($\delta^{13}C_{\text{wax}}$) isotope values of the n-C₃₁ alkane were measured and used to infer rainfall isotope changes and vegetation sources, respectively (Supplementary Tables S2–S4).” (lines 94–96). We have further clarified in the figure captions that δD_{wax} values are the δD values of the n-C₃₁ alkane.

Line 408, Fig 1 map needs elements of lat and long.

Thank you, we have changed the map accordingly.

REVIEWER COMMENTS

Reviewer #1 (Remarks to the Author):

I have read the authors rebuttal to my comments on a previous version and found their answers to be for the most part satisfying. The removal of the dD-gradient concept, which was clearly a problem with all of the reviewers, and a more straightforward interpretation of their dD records in terms of moisture source variations are good steps in the right direction. I don't think the authors understood my last comment, however, regarding the use of precession during the LGM and the current warm period. I noticed that their new explanation is more nuanced, and invokes differences in obliquity versus precession but the point I was trying to make was that these are not the only forcing factors that could explain their records. Changes in Pacific SSTs and/or El Niño-like or La Niña-like changes have also been invoked to explain past climate change along western South America and the authors did little to discuss these alternative hypotheses. Further annotations are included in the attached pdf file.

Reviewer #2 (Remarks to the Author):

Overview: The authors did a good job of addressing my comments from the first round of review. I appreciate that the dDwax records are now interpreted separately and that distinct drivers for each record are discussed. The Discussion is much improved over the previous draft.

I only have a few additional comments that would amount to minor revisions.

1) I appreciate that this study demonstrates how dDwax can trace moisture sources in mid-latitude Chile, but it is not the first to use dDwax in this way. A good example of this is Bhattacharya et al., 2018 Nature Geoscience (<https://doi.org/10.1038/s41561-018-0220-7>) which used dDwax as a tracer for moisture source for summer and winter precipitation in the western United States. These two sources of precipitation have distinct sources and thus distinct isotope signatures. Mentioning this previous work in the introduction (Line 69) would do justice to the fact that dDwax has been used as a moisture source indicator elsewhere.

2) I liked the authors' discussion of the existing pollen records in the Reviewer Response document and whether or not they would indicate a vegetation influence on dDwax (response to my comment on Line 101 in the previous draft). A shortened version of this should be included in the main text around Line 109 so as to acknowledge that d13Cwax doesn't provide a complete picture of vegetation change, because it only detects C4 presence and C4 isn't a big component of the ecosystems here.

Specific comments:

Line 26: extratropics

Line 61: evidence

Line 123: not sure how the p-value is computed here, but it needs to account for serial correlation and the reduced degrees of freedom. Or, you could jettison reporting the p-value

as an r of 0.8 is strong and the point is just that the two records are very similar.

Line 131: Be precise about what is seen b/t 12-8 ka. The 30S site indicates drier conditions/more enriched rainfall, but the 41S site does not. It shows relatively constant or even decreasing δD (more depleted values).

Line 135: Again, be precise. There is a change at 30S b/t 8-5 ka, there is a decreasing trend that is part of the full Holocene trend.

Line 144: see note above about p-values - you need to adjust for serial correlation of the time series if you are going to report these, which will make those p-values larger. A Breakdown of this issue and some suggestions for how to deal with it can be found at this link: http://seismo.berkeley.edu/~kirchner/eps_120/Toolkits/Toolkit_11.pdf. I personally prefer to use the Ebisuzaki method: [https://doi.org/10.1175/1520-0442\(1997\)010<2147:AMTETS>2.0.CO;2](https://doi.org/10.1175/1520-0442(1997)010<2147:AMTETS>2.0.CO;2). An implementation of this is available in the Pyleoclim software package: <https://pyleoclim-util.readthedocs.io/en/latest/>.

Reviewer #3 (Remarks to the Author):

I noticed that the authors have done a significant rewrite in this revised manuscript. My previous major concern is about the $\Delta\delta D_{wax}$ and the relevant results/discussions have been fully removed, so I'm okay with this edit. Regarding this revision, my concerns are:

The new Fig. 3 needs more careful discussions. In the current revision, the description and discussion regarding Fig. 3 pop up at Line 128 without enough context. I think I'm confused because I didn't see a clear reason why this part of the records needs to be singled out from Fig. 4. Also, some description seems to be wrong. For example, "...with relatively increased δD_p (should be δD_{wax} ?) values at 30°S and 41°S, ... (Line 131)" By eyeballing this figure, I feel the values are decreasing from 12-8, not increasing. In Fig. 2, SST patterns from those two sites from 0-12 ka are very different. At 41°S, the SSTs rapidly decreased from 12-8 then showed a gradual cooling until today. At 30°S, the early Holocene SSTs are pretty flat and the SSTs started to decrease during the mid-Holocene. This information is not incorporated into the interpretation of both δD_{wax} records in Fig. 3, and I wonder if it would impact the interpretation of the δD_{wax} , especially for this time interval.

I'm not fully convinced that the δD_{wax} at 41°S is solely related to moisture source variability (Line 147). Yes, both SSTs are significantly correlated, which is not that surprising because both records show the G-IG pattern. Do the authors imply that the δD_{wax} records at both 30 and 41°S solely reflect moisture source variability? I feel some clarification and explanation is needed.

The current δD_{wax} records at 41°S are just ~45 ka and barely cover one obliquity cycle (Figs. 4a,b). I don't think there is a hard rule but feel this needs to be acknowledged in the text.

The use of δD_p and δD_{wax} is very confusing throughout the discussion section. These two notations are different and should not be used interchangeably.

Minor comments:

Line 85, the latitude of the Concepcion station is shown as 37°S but it is 38°S in Fig. 1

Reviewer #1 (Remarks to the Author):

I have read the authors rebuttal to my comments on a previous version and found their answers to be for the most part satisfying. The removal of the dD-gradient concept, which was clearly a problem with all of the reviewers, and a more straightforward interpretation of their dD records in terms of moisture source variations are good steps in the right direction. I don't think the authors understood my last comment, however, regarding the use of precession during the LGM and the current warm period. I noticed that their new explanation is more nuanced, and invokes differences in obliquity versus precession but the point I was trying to make was that these are not the only forcing factors that could explain their records. Changes in Pacific SSTs and/or El Niño-like or La Niña-like changes have also been invoked to explain past climate change along western South America and the authors did little to discuss these alternative hypotheses. Further annotations are included in the attached pdf file.

We are thankful to the reviewer for the positive review and the pertinent comments. Indeed, we agree that a discussion on changes in ENSO to explain changes in δD at a glacial/interglacial timescale would be helpful. Therefore, we have now added a new paragraph (lines 197-216 of the revised manuscript) discussing the potential influence of ENSO-like states on rainfall in central Chile at a precessional timescale: “The subtropical moisture source variability as recorded at 30°S suggests a close link to atmosphere-ocean changes in the tropical Pacific at precessional timescales. Climate models point to austral winter zonal SST gradient changes across the equatorial Pacific as the key element controlling rainfall in subtropical Chile for these timescales¹⁸. Consistent with the δD_{wax} record at 30°S, the proxy record at 27.5°S and two different climate models suggest increased rainfall in subtropical Chile when zonal SST gradients in the tropical Pacific were increased and the subtropical jet was stronger. These climate pattern related to precession changes resemble modern interannual variability during the El Niño Southern Oscillation (ENSO)^{64,65}. The more humid interval during precession maxima also include the LGM consistent with previous records suggesting La Niña-like states during this time interval⁶⁶. In contrast to the modern climate with enhanced rainfall in subtropical Chile during El Niño events, the paleo data at 27.5°S and 30°S indicate an opposite relation of long-term ENSO-like intervals, i.e., more moisture during La Niña-states. This finding suggests a variable response to ENSO forcing at different timescales and/or different overall climate background states, i.e. the last glacial and the Holocene. This is consistent with studies showing that ENSO activity was reduced during the last glacial^{67,68}, and therefore a different mean ENSO state might be expected. Most importantly, there is no consensus on ENSO intensity during the Holocene and the last glacial period, neither from proxy records nor model simulations⁶⁹. Therefore, while ENSO accounts for about half of the winter rainfall variance in the modern climate^{64,65}, its influence on rainfall in central Chile was most likely reduced during the last glacial time and dominated by long-term ENSO-like variations of subtropical Chilean rainfall responding differently to SST changes in the tropical Pacific.” The new citations were added in the reference list (Refs. 64–69).

Ford, H. L., Ravelo, A. C. & Polissar, P. J. Reduced El Niño–Southern Oscillation during the Last Glacial Maximum. Science 347, 255–258 (2015).

Garreaud, R. D., Vuille, M., Compagnucci, R., & Marengo, J. Present-day South American climate. Palaeogeography, Palaeoclimatology, Palaeoecology 28, 180–195 (2009).

Lamy, F., Chiang, J. C. H., Martínez-Méndez, G., Thierens, M., Arz, H. W., Bosmans, J., Hebbeln, D., Lambert, F., Lembke-Jene, L. & Stuut, J.-B. W. Precession modulation of the South Pacific westerly wind belt over the past million years. Proceedings of the National Academy of Sciences of the United States of America 116, 23455–23460 (2019).

Leduc, G., Vidal, L., Cartapanis, O. & Bard, E. Modes of eastern equatorial Pacific thermocline variability: Implications for ENSO dynamics over the last glacial period. Paleoceanography 24, PA3202 (2009).

Lu, Z., Liu, Z., Zhu, J. & Cobb, K. M. A Review of Paleo El Niño-Southern Oscillation. *Atmosphere* 9, 130 (2018).

Montecinos, A. & Aceituno, P. Seasonality of the ENSO-related rainfall variability in central Chile and associated circulation anomalies. *Journal of Climate* 16, 281–296 (2003).

Tudhope, A. W., Chilcott, C. P., McCulloch, M. T., Cook, E. R., Chappell, J., Ellam, R. M., Lea, D. W., Lough, J.M. & Shimmield, G. B. Variability in the El Niño-Southern Oscillation through a glacial-interglacial cycle. *Science* 291, 1511–1517 (2001).

We are thankful for the annotations provided in the PDF file. Most of the comments and suggestions from the PDF have been accepted and included in the revised version of the manuscript (note that the line numbers refer here to the annotated PDF):

Line 25: We have changed “modulator” with “factor”.

Line 34: We have changed this wording into “strong seasonal latitudinal shifts”.

Line 43: Changed.

Line 51: Changed into “control almost all continental rainfall”.

Line 61: Corrected.

Line 83: We replaced “In order to evaluate ...” with “To assess ...”.

Line 126: “mainly” was removed.

Line 128: We have replaced “early Holocene” with “at the beginning of the Holocene”.

Line 129: Changed.

Line 133: Changed as suggested.

Lines 144-145: In agreement with a comment from another reviewer, this sentence has been rewritten into (lines 154-156 of the revised manuscript): “The SST records at 30°S and 41°S are positively correlated, as are the δD_{wax} records. However, an r value of 0.9 indicates that consistent SST variations occur along the Chilean margin, while a value of 0.6 suggests that moisture changes may have reacted to different forcing mechanisms at each latitude.”

Line 163-164: “While strongly” has been removed and “yet” has been added. Note that these sentences have been partly rewritten to improve their clarity.

Line 202: A summary with sampling interval in years and associated uncertainty of the cores is provided in the supplementary tables, not only for the age models, but also for the analysed samples.

Reviewer #2 (Remarks to the Author):

Overview: The authors did a good job of addressing my comments from the first round of review. I appreciate that the dD_{wax} records are now interpreted separately and that distinct drivers for each record are discussed. The Discussion is much improved over the previous draft.

I only have a few additional comments that would amount to minor revisions.

1) I appreciate that this study demonstrates how dD_{wax} can trace moisture sources in mid-latitude Chile, but it is not the first to use dD_{wax} in this way. A good example of this is Bhattacharya et al., 2018 *Nature Geoscience* (<https://doi.org/10.1038/s41561-018-0220-7>) which used dD_{wax} as a tracer for moisture source for summer and winter precipitation in the western United States. These two sources of precipitation have distinct sources and thus distinct isotope signatures. Mentioning this

previous work in the introduction (Line 69) would do justice to the fact that δD_{wax} has been used as a moisture source indicator elsewhere.

We much appreciated the comments of the reviewer. Unfortunately, we omitted citing the publication by Bhattacharya et al. This study is now cited (line 69) and the reference added in the reference list (Ref. 35).

2) I liked the authors' discussion of the existing pollen records in the Reviewer Response document and whether or not they would indicate a vegetation influence on δD_{wax} (response to my comment on Line 101 in the previous draft). A shortened version of this should be included in the main text around Line 109 so as to acknowledge that $\delta^{13}C_{wax}$ doesn't provide a complete picture of vegetation change, because it only detects C_4 presence and C_4 isn't a big component of the ecosystems here.

Following this advice, we have added a few sentences about the possible influence of vegetation on the δD_{wax} records (lines 112-116): “Indeed, as inferred from pollen records the regional vegetation on the adjacent land at the locations of both cores was dominated by C_3 plants (trees and woodland) during the whole time interval^{39,40,41}. Only during the deglaciation and the early Holocene grasses and shrubs increased, but C_4 monocotyledonous plants remain in low proportion (< 20%).” The references have been added to the reference list (Refs. 39-41).

Heusser, C. J. Ice age vegetation and climate of subtropical Chile. Palaeogeography, Palaeoclimatology, Palaeoecology 80, 107–127 (1990).

Valero-Garcés, B. L., Jenny, B., Rondanelli, M., Delgado-Huertas, A., Burns, S. J., Veit, H. & Moreno, A. Palaeohydrology of Laguna de Tagua Tagua (34°30'S) and moisture fluctuations in Central Chile for the last 46000 yr. Journal of Quaternary Science 20, 625–641 (2005).

Heusser, L., Heusser, C. J. & Piasias, N. Vegetation and climate dynamics of southern Chile during the past 50,000 years: results of ODP Site 1233 pollen analysis. Quaternary Science Reviews 25, 474–485 (2006).

Specific comments:

Line 26: extratropics

This has been changed (line 26).

Line 61: evidence

This has been changed and we made the verb agree with the subject (lines 61-62).

Line 123: not sure how the p-value is computed here, but it needs to account for serial correlation and the reduced degrees of freedom. Or, you could jettison reporting the p-value as an r of 0.8 is strong and the point is just that the two records are very similar.

Following the reviewer suggestion, we are now mentioning only the r value (line 127).

Line 131: Be precise about what is seen b/t 12-8 ka. The 30S site indicates drier conditions/more enriched rainfall, but the 41S site does not. It shows relatively constant or even decreasing δD (more depleted values).

We are thankful for this comment. Our sentence was not precise enough. What was meant here is that δD values are increased at both 30°S and 41°S during the early Holocene relative to values of the last deglaciation. To be clearer, we have now slightly changed the sentence into (lines 136-139): “This pattern is presumably related to a southward contraction of the SWW¹⁰, which is in close agreement with relatively increased δD_{wax} values at 30°S and 41°S compared to the deglaciation (Fig. 2A), suggesting a weaker influence of moisture from the high latitudes (Fig. 3B).”

Line 135: Again, be precise. There is a change at 30S b/t 8-5 ka, there is a decreasing trend that is part of the full Holocene trend.

We agree with this comment and we have now introduced a possible, but not significant, influence of temperature on the δD_{wax} values. We have slightly modified the two sentences into (lines 141-145): “This trend is reflected by a 6-7‰ increase in δD_{wax} values at 41°S suggesting a weaker influence of moisture from the high latitudes, although part of the D-enrichment (1‰) may be related to the concurrent 1°C SST warming (Fig. 2B). At 30°S, despite a slight decreasing trend (<2‰), which may be related to the <1°C SST cooling, the δD_{wax} values remained comparatively constant.”

Line 144: see note above about p-values - you need to adjust for serial correlation of the time series if you are going to report these, which will make those p-values larger. A Breakdown of this issue and some suggestions for how to deal with it can be found at this link:

http://seismo.berkeley.edu/~kirchner/eps_120/Toolkits/Toolkit_11.pdf. I personally prefer to use the Ebisuzaki method: [https://doi.org/10.1175/1520-0442\(1997\)010<2147:AMTETS>2.0.CO;2](https://doi.org/10.1175/1520-0442(1997)010<2147:AMTETS>2.0.CO;2). An implementation of this is available in the Pyleoclim software package: <https://pyleoclim-util.readthedocs.io/en/latest/>.

Here as well we are now mentioning only the r values (lines 155-156) following the reviewer’s suggestion.

Reviewer #3 (Remarks to the Author):

I noticed that the authors have done a significant rewrite in this revised manuscript. My previous major concern is about the $\Delta\delta D_{wax}$ and the relevant results/discussions have been fully removed, so I’m okay with this edit. Regarding this revision, my concerns are:

The new Fig. 3 needs more careful discussions. In the current revision, the description and discussion regarding Fig. 3 pop up at Line 128 without enough context. I think I’m confused because I didn’t see a clear reason why this part of the records needs to be singled out from Fig. 4. Also, some description seems to be wrong. For example, “...with relatively increased δD_p (should be δD_{wax} ?) values at 30°S and 41°S, ... (Line 131)” By eyeballing this figure, I feel the values are decreasing from 12-8, not increasing. In Fig. 2, SST patterns from those two sites from 0-12 ka are very different. At 41°S, the SSTs rapidly decreased from 12-8 then showed a gradual cooling until today. At 30°S, the early Holocene SSTs are pretty flat and the SSTs started to decrease during the mid-Holocene. This information is not incorporated into the interpretation of both δD_{wax} records in Fig. 3, and I wonder if it would impact the interpretation of the δD_{wax} , especially for this time interval.

We are thankful to the reviewer for the positive review and especially to this comment, which is partly similar to one of Reviewer #2 comment. A focus on the Holocene has been made based on comments from the first round of revision. As a large set of paleoenvironmental records are available for the Holocene compared to the last glacial and deglaciation, the Holocene part of the record can be used as a “proof of concept” that δD_{wax} records off south-central Chile are indeed reflecting changes in the high versus low latitude sources of moisture during the Holocene. To make this aspect clearer, we have now introduced three headlines structuring the Discussion:

Origin of the δD_{wax} signal (line 109)

δD_{wax} reflects changes in the moisture source during the Holocene (line 133)

Implications for rainfall modulation in coastal Chile over the past 50 kyr (line 153)

Concerning the description and discussion of Fig. 3, we fully agree that they were not clear. What was meant in the paragraph about the early Holocene is that δD_{wax} values are increased at both 30°S and 41°S during the early Holocene relative to values of the last deglaciation. To be clearer, we have now

slightly changed the sentence into (lines 136-139): “This pattern is presumably related to a southward contraction of the SWW¹⁰, what is in close agreement with relatively increased δD_{wax} values at 30°S and 41°S compared to the deglaciation (Fig. 2A), suggesting a weaker influence of moisture from the high latitudes (Fig. 3B).” Concerning the mid-Holocene (8-5 ka), we have now partly rewritten this paragraph and introduced a possible, but not significant, influence of temperature on Holocene δD_{wax} values as suggested by the reviewer (lines 140-146): “The middle Holocene (8-5 ka) was characterized by a further drying trend in south-central Chile^{45,48,52,54,55}, as well as south of 50°S^{10,11}. This trend is reflected by a 6-7‰ increase in δD_{wax} values at 41°S suggesting a weaker influence of moisture from the high latitudes, although part of the D-enrichment (1‰) may be related to the concurrent 1°C SST warming (Fig. 2B). At 30°S, despite a slight decreasing trend (<2‰), which may be related to the <1°C SST cooling, the δD_{wax} values remained comparatively constant. A further poleward shift of the SWW combined with a decreasing wind strength may explain drier conditions in central and southern Chile during this time-interval.”

I’m not fully convinced that the δD_{wax} at 41°S is solely related to moisture source variability (Line 147). Yes, both SSTs are significantly correlated, which is not that surprising because both records show the G-IG pattern. Do the authors imply that the δD_{wax} records at both 30 and 41°S solely reflect moisture source variability? I feel some clarification and explanation is needed.

Thank you for this comment. We imply here that the δD_{wax} record reflects mainly moisture source variability, but a temperature effect cannot be completely ruled out. Therefore, we are now introducing a potential minor effect of temperature on δD_{wax} values not only for the Holocene (“However, a minor effect of temperature on δD_{wax} values cannot be completely ruled out.”; lines 151-152), but also for the glacial and deglacial part of the records: “However, temperature may also affect δD_{wax} variability to a minor extent as suggested from the Holocene part of the record.” (lines 158-159). And we have tempered our words in the sentence in question: “Changes in δD_{wax} values at 41°S are most likely related mainly to moisture source variability as the amount effect is restricted to lower, subtropical latitudes^{43,44}” (lines 157-158).

The current δD_{wax} records at 41°S are just ~45 ka and barely cover one obliquity cycle (Figs. 4a,b). I don’t think there is a hard rule but feel this needs to be acknowledged in the text.

We agree and we are now mentioning that “Therefore, although the δD_{wax} record is restricted to the last 45 kyr, it appears that changes in the moisture source at 41°S, and thus the location and strength of the SWW, are modulated by obliquity.” (lines 169-171).

The use of δD_p and δD_{wax} is very confusing throughout the discussion section. These two notations are different and should not be used interchangeably.

We are sorry for this. For clarity and unambiguity, we decided to use only δD_{wax} throughout the manuscript, knowing that δD_{wax} reflects rainfall δD (line 120).

Minor comments:

Line 85, the latitude of the Concepcion station is shown as 37°S but it is 38°S in Fig. 1

The latitude of Concepcion is ca. 37°S. This has been corrected in Fig. 1 and in the figure caption.

REVIEWERS' COMMENTS

Reviewer #1 (Remarks to the Author):

I have reviewed this re-revised version of the manuscript and I am satisfied with the authors responses. Their addition in the discussion of the many different ways in which ENSO can impact the climate system on different timescales is particularly interesting. I have no further comments or revisions to the manuscript itself.

Reviewer #2 (Remarks to the Author):

I am satisfied with the last round of revisions made by the authors.

Reviewer #3 (Remarks to the Author):

The authors have addressed all my concerns. I find this manuscript much improved and ready for publication.

REVIEWERS' COMMENTS

Reviewer #1 (Remarks to the Author):

I have reviewed this re-revised version of the manuscript and I am satisfied with the authors responses. Their addition in the discussion of the many different ways in which ENSO can impact the climate system on different timescales is particularly interesting. I have no further comments or revisions to the manuscript itself.

Reviewer #2 (Remarks to the Author):

I am satisfied with the last round of revisions made by the authors.

Reviewer #3 (Remarks to the Author):

The authors have addressed all my concerns. I find this manuscript much improved and ready for publication.

We are thankful to the reviewers for all the affords and comments to improve the manuscript. We are happy to know that that they are satisfied with the last round of revisions and that they consider our manuscript ready for publication.